# Beyond Intuition:
# Rethinking Token Attributions inside Transformers

**Jiamin Chen**                                    *jiaminchen@buaa.edu.cn*
*Beihang University & Baidu Inc.*

**Xuhong Li**                                    *lixuhong@baidu.com*
*Baidu Inc.*

**Lei Yu**                                    *yulei@buaa.edu.cn*
*Beihang University & Beihang Hangzhou Innovation Institute Yuhang*

**Dejing Dou**                                    *doudejing@baidu.com*
*Baidu Inc.*

**Haoyi Xiong**                                    *xionghaoyi@baidu.com*
*Baidu Inc.*

**Reviewed on OpenReview:** *https://openreview.net/forum?id=rm0zIzlhcX*

## Abstract

The multi-head attention mechanism, or rather the Transformer-based models have always been under the spotlight, not only in the domain of text processing, but also for computer vision. Several works have recently been proposed around exploring the token attributions along the intrinsic decision process. However, the ambiguity of the expression formulation can lead to an accumulation of error, which makes the interpretation less trustworthy and less applicable to different variants. In this work, we propose a novel method[1] to approximate token contributions inside Transformers. We start from the partial derivative to each token, divide the interpretation process into *attention perception* and *reasoning feedback* with the chain rule and explore each part individually with explicit mathematical derivations. In attention perception, we propose the head-wise and token-wise approximations in order to learn how the tokens interact to form the pooled vector. As for reasoning feedback, we adopt a noise-decreasing strategy by applying the integrated gradients to the last attention map. Our method is further validated qualitatively and quantitatively through the faithfulness evaluations across different settings: single modality (BERT and ViT) and bi-modality (CLIP), different model sizes (ViT-L) and different pooling strategies (ViT-MAE) to demonstrate the broad applicability and clear improvements over existing methods.

## 1 Introduction

The Transformer (Vaswani et al., 2017) and its variants (Dosovitskiy et al., 2021; Devlin et al., 2019; Brown et al., 2020) take a leading place in the domain of natural language processing (NLP), where texts are firstly tokenized into words or sub-words, identified as tokens, and then fed into the deep neural network. The main architecture of Transformer consists of several attention blocks, where the query-relevant tokens are captured and combined to make a new representation. With its huge success in NLP, more and more Transformer-based models have been proposed for multiple tasks, such as image classification (Dosovitskiy et al., 2021), object detection (Carion et al., 2020), VQA (Antol et al., 2015) and GQA (Hudson & Manning, 2019). For images, tokens are usually the patches which are uniformly segmented (Dosovitskiy et al., 2021), obtained with a shift window (Liu et al., 2021) or captured with an object detector (Tan & Bansal, 2019; Li et al., 2019).

---

[1]Code available at https://github.com/jiaminchen-1031/transformerinterp and InterpretDL (Li et al., 2022) as well.

Figure 1: With different ViT variants, we observe an accumulation of error for the intuitive interpretation method Generic Attribution when the model gets deeper or adopts a global pooling strategy, which makes the results contradictory with the model performances.

Explaining how tokens are mixed and used inside the Transformer to make the final prediction can help a lot in understanding, debugging and refining the model. Some classic interpretation methods can be applied with modifications considering a Transformer literature, such as the input gradient based methods (Selvaraju et al., 2017; Sundararajan et al., 2017; Smilkov et al., 2017). Regardless of their versatility for all differentiable models, the explanation results can be noisy to certain extent, due to the saturation and vanishing of gradients caused by numerous non-linear components in the deep models. Some attention-based explanation methods have also been proposed for the Transformers (Michel et al., 2019; Abnar & Zuidema, 2020; Chefer et al., 2021a;b; Hao et al., 2021), despite the disputation about the legitimacy of attentions being an explanation (Jain & Wallace, 2019; Wiegreffe & Pinter, 2019; Kobayashi et al., 2020). With more characteristics inside Transformer taken into account, these attention-based algorithms generally provide more faithful explanation results than simply adopting the raw attentions.

However, existing attention-based Transformer explanation methods are built either on too strong assumptions, or without clear theoretical frameworks. Rollout (Abnar & Zuidema, 2020) takes skip connections into consideration but ignores the existing effects of linear projections inside attention blocks (as detailed in Section 3.2). Transformer Attribution (Chefer et al., 2021a) and Generic Attribution (Chefer et al., 2021b) combine the gradients with layer-wise relevance propagation (Binder et al., 2016) (LRP) or attention maps along a rolling out path, and eliminate the negative components in each attention block. Though gradients in a way reflect the direct influence to the final objective, involving them intuitively and masking the negatives in each attention block risk losing useful information from relevance scores or attention maps.

In this work, we intend to bring up the potential risks of introducing the methods intuitively. The unidentified errors from intuitive designs can be disclosed and magnified under different settings, causing the methods with good performances in the vanilla models fail in explaining their variants. We show in Figure 1 some visualizations with Generic Attribution to different types of Vision Transformers (Dosovitskiy et al., 2021; He et al., 2022). In spite of its good performances in explaining ViT-Base, the visualizations on other ViT variants show some contradictions to the models' properties, where the model with better performances (ViT-L) is visualized with a focus on less semantic pixels, and the explanation results do not correspond with the semantics enhancement brought by the masked pretraining technique (ViT-MAE). These contradictions decrease the trustworthiness of interpretation methods and make them less applicable to all Transformer-based models.

**Contributions.** In order to explicitly explain how the model predicts and theoretically formalize the token interactions inside Transformers, we introduce a novel explanation method with main contributions as following:

**1)** Based on the partial derivative of the loss function to each token, we propose a novel interpretation framework with the chain rule, dividing the whole process into the *Attention Perception* and *Reasoning Feedback*, as shown in Figure 2. We believe that this framework can inspire more interpretation methods with other approximation strategies.

**2)** In attention perception, we look into the relation between the input and output in each attention block and mathematically derive two recurrence formulas with our head-wise and token-wise attention maps, where the token-wise one achieves $20\times$ more accurate on average than Attention Rollout. By recurring from the first to the last attention block, we reach the expression indicating the contributions of input tokens in the pooled vector before prediction head.

**3)** To validate the faithfulness of the proposed explanation algorithm, we follow the previous evaluation methods (Samek et al., 2017; Vu et al., 2019; DeYoung et al., 2020), and compare ours with existing algorithms. The evaluation settings cover different modalities, different sizes and different pooling strategies concerning Transformers, including BERT-Base (Devlin et al., 2019) for texts, ViT-Base (Dosovitskiy et al., 2021) for images, CLIP (Radford et al., 2021) for bi-modality, ViT-Large (Dosovitskiy et al., 2021) for the large model, and ViT-MAE (He et al., 2022) for the global pooling strategy. Our approach outperforms other strong baselines (e.g., (Abnar & Zuidema, 2020; Chefer et al., 2021a;b)) through quantitative metrics and qualitative visualizations, and shows better applicability to various settings. More analyses and ablation studies are also provided.

## 2 Related work

**Transformer-based Models.** For the first time being proposed by Vaswani et al. (2017), the Transformer has been rapidly and widely used for almost all kinds of deep learning tasks. A variety of derived methods (Dosovitskiy et al., 2021; Devlin et al., 2019; Brown et al., 2020; Liu et al., 2019) has sprung up in recent years. These Transformer-based models have become the state of the art in most of the NLP benchmarks and stepped in the domain of computer vision as well (Dosovitskiy et al., 2021; Liu et al., 2021). Vision Transformer (ViT) (Dosovitskiy et al., 2021) takes a sequence of image patches, regraded as tokens, into successive attention blocks and feeds the last class token [CLS] into prediction head for downstream tasks. With its broad usage covering both image and text, the Transformer architecture has been explored for a lot of bi-modal models, e.g., VisualBERT (Li et al., 2019), LXMERT (Tan & Bansal, 2019), CLIP (Radford et al., 2021) and VLM (Xu et al., 2021). A single or dual self-attention/cross-attention based encoder is used to get the representations of both text and image. Thus, being capable to explain the decision process of these Transformer-based models is rather important for a deeper understanding and a refinement for better performance.

**Explainability of Deep Models.** Deep models are well known for their superb performance as well as their black-box nature. Before Transformers, many explanation methods have been proposed to explain the deep models. For instance, as a post-hoc local explanation method, LIME (Ribeiro et al., 2016) locally explains the deep model at a single data point. Many methods based on gradients have also been proposed because of the differentiability of most deep models. Smooth Gradient (SG) (Smilkov et al., 2017) and Integrated Gradient (IG) (Sundararajan et al., 2017) are two explanation algorithms to produce input gradient based saliency maps, through adding noisy inputs, or using the integral respectively. More methods focus on specific model structures, e.g., CAM/Grad-CAM (Zhou et al., 2016; Selvaraju et al., 2017; Wang et al., 2020) for convolutional networks, GNNExplainer (Ying et al., 2019) for graph neural networks, and GAN dissection (Bau et al., 2019) for generative adversarial networks.

**Explainability for Transformers.** Some Transformer-specific explanation methods have also been proposed recently, especially towards the goal of better understanding the characteristic components inside. Previous studies (Bahdanau et al., 2015; Xu et al., 2015; Choi et al., 2016) show that raw attention maps can be used as explanations for attention-based models. Considering the multi-head mechanism, Michel et al. (2019) find that the contributions of heads vary a lot from one to another, and therefore propose that pruning the unimportant heads has little impact to the model. Attention Rollout (Abnar & Zuidema, 2020) assumes that the input tokens are linearly combined based on the attention weights and takes the [CLS] token as explanations. Generic Attribution (Chefer et al., 2021b) generalizes the idea of Rollout and adds the gradient information to each attention map, while Transformer Attribution (Chefer et al., 2021a) exploits LRP (Binder et al., 2016) and gradients together for getting the explanations.

## 3 Method

This section introduces the explanation expression we propose for Transformers. To facilitate reading, we present in Figure 2 a roadmap to illustrate the main idea of our proposed method.

**Roadmap.** We consider the token attributions inside Transformers as the derivative of the loss function to the input tokens. In Section 3.1, by proposing a novel basis $\mathbb{B} = \{\tilde{X}_{\text{CLS}}, \tilde{X}_1, \ldots, \tilde{X}_N\}$, where each vector represents the corresponding token, we can directly study the importance of token $i$ as $\frac{\partial \mathcal{L}^c(X)}{\partial t_i}$ with $(X)_{\mathbb{B}} = (t_0, \ldots, t_N)$. The basis vectors $\tilde{X}_i$ are identified in Section 3.2 by unfolding the successive attention blocks inside Transformers. Applying the chain rule with the output of last attention block $\mathbf{Z}^{(L)}$, we divide the derivative into: *Attention Perception* $\mathbf{P}^{(L)} := \frac{\partial \mathbf{Z}^{(L)}}{\partial t_i}$ and *Reasoning Feedback* $\mathbf{F}^c := \frac{\partial \mathcal{L}^c(X)}{\partial \mathbf{Z}^{(L)}}$, and combine them to have the expression of our explanation method.

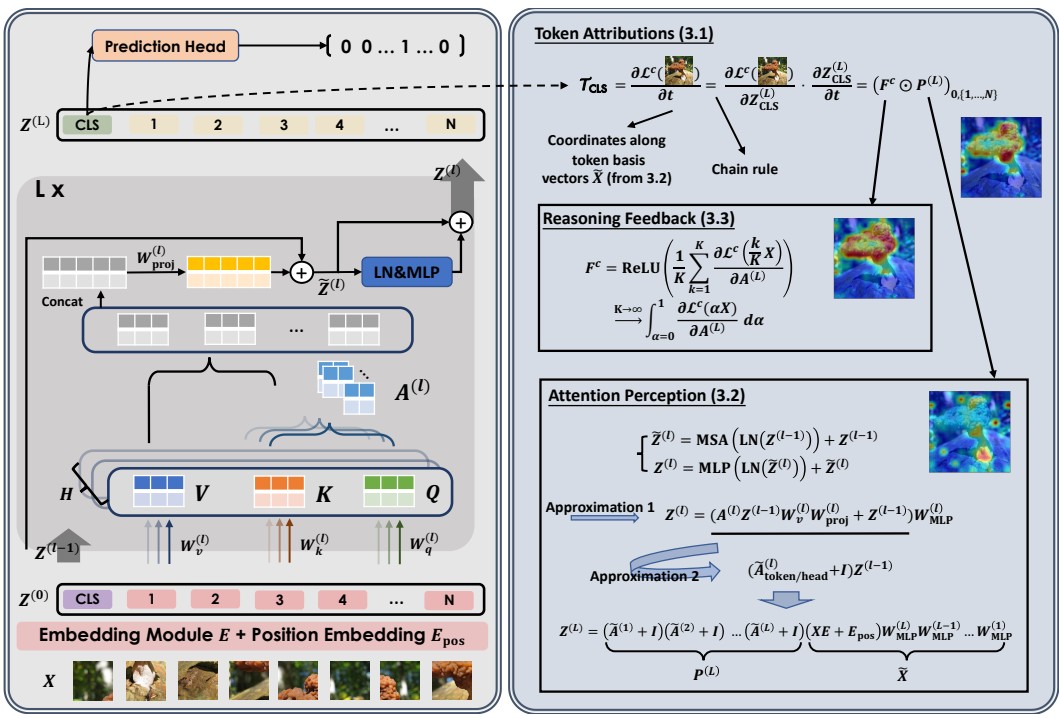

Figure 2: Illustration of our method with ViT (Dosovitskiy et al., 2021). We show in the *left* how information propagates inside ViT. For our method in the *right*, we start from the partial derivative, divide the whole process into *Attention Perception* and *Reasoning Feedback* with the chain rule, and analyze each sub-process to obtain the final expression.

In Section 3.2, to obtain $\mathbf{P}^{(L)}$, we focus on deriving a recursive relation between the input $\mathbf{Z}^{(l-1)}$ and output $\mathbf{Z}^{(l)}$ in attention block $l$. With Approximation 1 and Approximation 2, we derive a recursive relation $\mathbf{Z}^{(l)} \approx (\tilde{\mathbf{A}}^{(l)} + I)\mathbf{Z}^{(l-1)}W_{\mathrm{MLP}}^{(l)}$ and propose our head-wise ($\tilde{\mathbf{A}}_{\mathrm{head}}^{(l)}$) and token-wise ($\tilde{\mathbf{A}}_{\mathrm{token}}^{(l)}$) attention maps. This recursion eventually leads to $\mathbf{Z}^{(L)} = \mathbf{P}^{(L)}\tilde{X}$, which reveals both $\mathbb{B}$ and $\mathbf{P}^{(L)}$.

In Section 3.3 for $\mathbf{F}^c$, explaining the process of passing the pooled vector for the final prediction, we introduce the technique of integration to decrease the possible noise on irrelevant tokens.

## Preliminaries

Before starting the introduction of our method, in this subsection, we recall several important concepts to prepare for the following derivation.

**Transformers.** We present in the *left* panel of Figure 2 a Transformer-based model consisting of $L$ blocks. The input $X$ for the Transformer is a sequence of $N$ tokens. After the embedding module $\mathbf{E}$, the tokens are turned into vectors of dimension $D$, with a special [CLS] token at the top and a positional embedding $\mathbf{E}_{\mathrm{pos}} = [\mathbf{E}_{\mathrm{pos},0}; \mathbf{E}_{\mathrm{pos},1}; \ldots; \mathbf{E}_{\mathrm{pos,N}}]$. Let $l$ be the $l$-th attention block composed of layer normalisation, multi-head self-attention (MSA), skip connections and projection layers. We denote the input of the block $l$ as $\mathbf{Z}^{(l-1)}$, and the output $\mathbf{Z}^{(l)}$. In the MSA module, the attentions $\mathbf{A}^{(l)}$, defined as $\mathrm{softmax}(Q \cdot K^{\mathrm{T}}/\sqrt{\mathrm{d_h}})$, capture the important tokens from a certain query and combines them to make an output. Here for self-attention, query (Q), key (K) and value (V) are from trainable transformations of the same input data. Layer Normalisation (LN) (Ba et al., 2016) re-scales the input for a single training case. The MLP module consists of two linear layers with an activation function ReLU/GeLU in between. In brief, we have

$$\mathbf{Z}^{(0)} = [X_{\mathrm{CLS}}; X_1\mathbf{E}; X_2\mathbf{E}; ...; X_N\mathbf{E}] + \mathbf{E}_{\mathrm{pos}}, \qquad \mathbf{E} \in \mathbb{R}^{(d_X)\times D}, \mathbf{E}_{\mathrm{pos}} \in \mathbb{R}^{(N+1)\times D} \quad , \qquad (1)$$

$$\tilde{\mathbf{Z}}^{(l)} = \mathrm{MSA}(\mathrm{LN}(\mathbf{Z}^{(l-1)})) + \mathbf{Z}^{(l-1)}, \qquad\qquad l = 1, ..., L \quad , \qquad (2)$$

$$\mathbf{Z}^{(l)} = \mathrm{MLP}(\mathrm{LN}(\tilde{\mathbf{Z}}^{(l)})) + \tilde{\mathbf{Z}}^{(l)}, \qquad\qquad l = 1, ..., L \quad . \qquad (3)$$

Note that the MSA module operates on the sub-spaces of dimension $D_h$, where $h$ refers to the head index, and we have $H$ heads so that $HD_h = D$.

Most Transformers perform the classification task via the last [CLS] embedding, while some use global pooling instead (He et al., 2022). In this section, we introduce our method particularly with [CLS] pooling, but the same methodology can also be applied for other cases.

**Coordinates and Partial Derivative.** Let $\mathbb{B} = \{V_1, V_2, \ldots, V_n\}$ be a basis of a vector space $V$ with finite dimension $n$. Given a vector $Z \in V$, let $(a_1, \ldots, a_n)$ be the coordinates of $Z$ over $\mathbb{B}$; that is $Z = \sum_{i=1}^{n} a_i V_i$. We denote $(Z)_{\mathbb{B}} = (a_1, \ldots, a_n)$.

Setting $f : V \to \mathbb{R}$ a function. For $y = f(Z) = f(a_1, \ldots, a_n)$, the partial derivative of $f$ with respect to the variable $a_i$ is denoted as $\frac{\partial y}{\partial a_i} = \frac{\partial f(a_1, a_2, \cdots, a_n)}{\partial a_i}$.

In the derivation of our method, we use $\mathbf{A}_{i,j}$ to represent the $(i+1)$-th row and $(j+1)$-th column of the matrix $\mathbf{A}$, $\mathbf{A}_{i,\cdot}$ for the entire $(i+1)$-th row of the matrix $\mathbf{A}$ and $\mathbf{A}_{i,\{j,\ldots,k\}}$ for the $(i+1)$-th row and $(j+1)$-th to $(k+1)$-th columns of the matrix $\mathbf{A}$. For more details, we provided a full table of notations in the appendix.

### 3.1 Token Attributions inside Transformers

Let $X$ be the input and $\mathcal{L}^c$ the loss function while predicting class $c$. The aim of our method is to obtain the importance of each input token with respect to the decision process. Using the back-propagated gradients to the input $\frac{\partial \mathcal{L}^c(X)}{\partial X}$ could be an answer, but here we would like to study directly the entire attribution of each token, since the transformation from the obtained high-dimensional explanation results to a single scalar which represents the contribution of each token can bring unexpected noise. By formulating the input-output relations in attention blocks (details in Section 3.2), we propose a set $\mathbb{B}$ of vectors,

$$\mathbb{B} = \{\tilde{X}_{\text{CLS}}, \tilde{X}_1, \ldots, \tilde{X}_N\} \quad , \tag{4}$$

where each element represents the corresponding token and will be identified later in Section 3.2.

In this way, we can rewrite the input vector $X$ as $(X)_{\mathbb{B}} = (t_0, t_1, \ldots, t_N)$, where $t_0$ is the coordinate over $\tilde{X}_{\text{CLS}}$ and $t_i$ is the coordinate over $\tilde{X}_i$ for $i \in \{1, \ldots, N\}$. Thus, we can define the attribution of the $i$-th token as $\frac{\partial \mathcal{L}^c(X)}{\partial t_i}$.

From the chain rule, we develop $\frac{\partial \mathcal{L}^c(X)}{\partial t_i}$ and denote $\mathbf{Z}_{\text{CLS}}^{(L)}$ the [CLS] token of the output in the last $L$-th attention block, having

$$\frac{\partial \mathcal{L}^c(X)}{\partial t_i} = \frac{\partial \mathcal{L}^c(X)}{\partial \mathbf{Z}_{\text{CLS}}^{(L)}} \cdot \frac{\partial \mathbf{Z}_{\text{CLS}}^{(L)}}{\partial t_i} = \frac{\partial \mathcal{L}^c(X)}{\partial (\mathbf{Z}_{\text{CLS}}^{(L)})_{\mathbb{B}}} \cdot \frac{\partial (\mathbf{Z}_{\text{CLS}}^{(L)})_{\mathbb{B}}}{\partial t_i} \quad . \tag{5}$$

With the analyses in Section 3.2, $\mathbf{Z}_{\text{CLS}}^{(L)}$ can be rewritten under the basis $\mathbb{B}$ as $(\mathbf{Z}_{\text{CLS}}^{(L)})_{\mathbb{B}} = \mathbf{P}_{0,\cdot}^{(L)}$, indicating the attributions of each input token to the last [CLS] before the prediction head. Here we make an approximation that $\frac{\partial \mathcal{L}^c(X)}{\partial \mathbf{P}_{0,\cdot}^{(L)}} \approx \frac{\partial \mathcal{L}^c(X)}{\partial \mathbf{A}_{0,\cdot}^{(L)}}$, which could be tenable for the cases where raw attentions can in a way represent the attributions in last [CLS] token. So we have $\frac{\partial \mathcal{L}^c(X)}{\partial (\mathbf{Z}_{\text{CLS}}^{(L)})_{\mathbb{B}}} \approx \frac{\partial \mathcal{L}^c(X)}{\partial \mathbf{A}_{0,\cdot}^{(L)}}$. As for the second term $\frac{\partial (\mathbf{Z}_{\text{CLS}}^{(L)})_{\mathbb{B}}}{\partial t_i}$, we can have $\frac{\partial (\mathbf{Z}_{\text{CLS}}^{(L)})_{\mathbb{B}}}{\partial t_i} = (0, \ldots, \mathbf{P}_{0,i}^{(L)}, \ldots, 0)$.

Now plugging the above formulas into Eq. (5), we can compute the attribution of $i$-th token, which is

$$\frac{\partial \mathcal{L}^c(X)}{\partial t_i} \approx \frac{\partial \mathcal{L}^c(X)}{\partial \mathbf{A}_{0,\cdot}^{(L)}} \cdot \frac{\partial (\mathbf{Z}_{\text{CLS}}^{(L)})_{\mathbb{B}}}{\partial t_i} = \frac{\partial \mathcal{L}^c(X)}{\partial \mathbf{A}_{0,i}^{(L)}} \mathbf{P}_{0,i}^{(L)} \quad . \tag{6}$$

Combining all the tokens together and using $\mathbf{F}^c$ to denote $\frac{\partial \mathcal{L}^c(X)}{\partial \mathbf{A}^{(L)}}$, we reach the expression of our explanation method:

$$\left( \frac{\partial \mathcal{L}^c(X)}{\partial t_1}, \ldots, \frac{\partial \mathcal{L}^c(X)}{\partial t_i}, \ldots, \frac{\partial \mathcal{L}^c(X)}{\partial t_N} \right) \approx \left( \mathbf{F}^c \odot \mathbf{P}^{(L)} \right)_{0, \{1 \ldots N\}} \quad . \tag{7}$$

**Our Explanation Method.** *For the Transformer-based model, the attributions of partitioned tokens to the final decision $\mathcal{T}$ can be expressed as a Hadamard product of two parts:*

$$\mathcal{T} = \left( \mathbf{P}^{(L)} \odot \mathbf{F}^c \right)_{\mathcal{P}} \quad . \tag{8}$$

We denote the first part $\mathbf{P}^{(L)}$, representing how the input tokens integrate in $L$ successive attention blocks and being regarded as a process of representation learning, so called *attention perception*. The second part, denoted as $\mathbf{F}^c$, implies how the last [CLS] token is used for predicting a certain class $c$, so called *reasoning feedback*. Their details will be introduced in Sections 3.2 and 3.3 respectively. The subscript $\mathcal{P} \in \{\text{cls}, \text{glp}, \dots\}$ is determined by the pooling strategy, such as [CLS] pooling and global pooling. In [CLS] pooling, it refers to the indices $\{(0,1), \dots, (0, N)\}$ of the matrix $\mathcal{T}_{\text{cls}} = (\mathbf{P}^{(L)} \odot \mathbf{F}^c)_{0, \{1 \dots N\}}$. In global pooling, by replacing $\mathbf{Z}_{\text{CLS}}^{(L)}$ with $[\mathbf{Z}_1^{(L)}; \dots; \mathbf{Z}_N^{(L)}]$, we obtain its expression $\mathcal{T}_{\text{glp}} = \sum_{k=1}^{k=N} (\mathbf{P}^{(L)} \odot \mathbf{F}^c)_{k, \{1 \dots N\}}$.

## 3.2 Attention Perception

In this subsection, in order to identify $\mathbf{P}^{(L)}$, we analyze how the input tokens interact to obtain the last [CLS] representation. We first unfold each attention block to derive the relation between the input and output of the block.

**Approximation 1.** *In MSA blocks with attentions $\mathbf{A}^{(l)}$, we approximate the relations between input $\mathbf{Z}^{(l-1)}$ and output $\mathbf{Z}^{(l)}$ as $\mathbf{Z}^{(l)} \approx (\mathbf{A}^{(l)} \mathbf{Z}^{(l-1)} W^{(l)} + \mathbf{Z}^{(l-1)}) W_{\text{MLP}}^{(l)}$.*

*Justification.* The non-linear components such as layer normalisation (LN) and activation function play an important role in deep neural networks. Here we make a linear assumption and neglect the effects of LN and ReLU/GeLU, because they operate on individual tokens and have slight influence on token interaction. Thus we simplify the information flow in the attention block as linear combinations and transformations for all the tokens:

$$\tilde{\mathbf{Z}}^{(l)} \overset{\text{linear}}{\underset{\text{assum.}}{=}} \mathbf{A}^{(l)} \cdot (\mathbf{Z}^{(l-1)} W_v^{(l)}) W_{proj}^{(l)} + b_{proj}^{(l)} + \mathbf{Z}^{(l-1)} \quad , \qquad l = 1, ..., L \quad , \tag{9}$$

$$\mathbf{Z}^{(l)} \overset{\text{linear}}{\underset{\text{assum.}}{=}} (\tilde{\mathbf{Z}}^{(l)} W_{\text{MLP},1}^{(l)} + b_{\text{MLP},1}^{(l)}) W_{\text{MLP},2}^{(l)} + b_{\text{MLP},2}^{(l)} + \tilde{\mathbf{Z}}^{(l)} \quad , \qquad l = 1, ..., L \quad , \tag{10}$$

where $W_v^{(l)}, W_{proj}^{(l)} \in \mathbb{R}^{D \times D}$ and $b_{proj}^{(l)} \in \mathbb{R}^D$ indicate the linear transformations inside Eq. (2), and $b_{\text{MLP},1}^{(l)} \in \mathbb{R}^{D_{\text{hidden}}}, b_{\text{MLP},2}^{(l)} \in \mathbb{R}^D$, $W_{\text{MLP},1}^{(l)} \in \mathbb{R}^{D \times D_{\text{hidden}}}, W_{\text{MLP},2}^{(l)} \in \mathbb{R}^{D_{\text{hidden}} \times D}$ in Eq. (3).

After putting Eq. (9) into Eq. (10), we rearrange the formula and obtain the relations between the input and output of the attention block $l$ as:

$$\mathbf{Z}^{(l)} = (\mathbf{A}^{(l)} \mathbf{Z}^{(l-1)} W^{(l)} + \mathbf{Z}^{(l-1)}) W_{\text{MLP}}^{(l)} + B^{(l)} \quad , \tag{11}$$

where $W_{\text{MLP}}^{(l)} = W_{\text{MLP},1}^{(l)} W_{\text{MLP},2}^{(l)} + I$, $W^{(l)} = W_v^{(l)} W_{proj}^{(l)}$ and $B^{(l)} = b_{proj}^{(l)} (W_{\text{MLP},1}^{(l)} W_{\text{MLP},2}^{(l)} + I) + b_{\text{MLP},1}^{(l)} W_{\text{MLP},2}^{(l)} + b_{\text{MLP},2}^{(l)}$. We will neglect $B^{(l)}$ because it is shared by all tokens and does not make tokens interacted. Thus, we obtain the relations that $\mathbf{Z}^{(l)} \approx (\mathbf{A}^{(l)} \mathbf{Z}^{(l-1)} W^{(l)} + \mathbf{Z}^{(l-1)}) W_{\text{MLP}}^{(l)}$.

**Approximation 2.** *If we can find $\tilde{\mathbf{A}}^{(l)}$ satisfying $\mathbf{A}^{(l)} \mathbf{Z}^{(l-1)} W^{(l)} + \mathbf{Z}^{(l-1)} \approx (\tilde{\mathbf{A}}^{(l)} + I) \mathbf{Z}^{(l-1)}$, the relations between the input tokens and the output vectors of last attention block $\mathbf{Z}^{(L)}$ can be written as $\mathbf{Z}^{(L)} \approx \mathbf{P}^{(L)} (\tilde{X}_{\text{CLS}}, \tilde{X}_1, \dots, \tilde{X}_N)^{\text{T}}$, where $\mathbf{P}^{(L)} = (\tilde{\mathbf{A}}^{(L)} + I) \dots (\tilde{\mathbf{A}}^{(1)} + I)$ and $\tilde{X}_i = (X_i \mathbf{E} + \mathbf{E}_{\text{pos,i}}) W_{\text{MLP}}^{(L)} \dots W_{\text{MLP}}^{(1)}$.*

*Justification.* With $\mathbf{A}^{(l)} \mathbf{Z}^{(l-1)} W^{(l)} + \mathbf{Z}^{(l-1)} \approx (\tilde{\mathbf{A}}^{(l)} + I) \mathbf{Z}^{(l-1)}$, we can obtain the recurrence relation $\mathbf{Z}^{(l)} = (\tilde{\mathbf{A}}^{(l)} + I) \mathbf{Z}^{(l-1)} W_{\text{MLP}}^{(l)}$. Hence, the conclusion can be reached directly from recursion. We recur from the last block $L$ to the first block and obtain the relation between output of block $L$ and input tokens:

$$\begin{pmatrix} \mathbf{Z}_{\text{CLS}}^{(L)} \\ \mathbf{Z}_1^{(L)} \\ \dots \\ \mathbf{Z}_N^{(L)} \end{pmatrix} \approx (\tilde{\mathbf{A}}^{(L)} + I) \dots (\tilde{\mathbf{A}}^{(1)} + I) \begin{pmatrix} (X_{\text{CLS}} + \mathbf{E}_{\text{pos,0}}) \tilde{\mathbf{W}} \\ (X_1 \mathbf{E} + \mathbf{E}_{\text{pos,1}}) \tilde{\mathbf{W}} \\ \dots \\ (X_N \mathbf{E} + \mathbf{E}_{\text{pos,N}}) \tilde{\mathbf{W}} \end{pmatrix} = \mathbf{P}^{(L)} \begin{pmatrix} \tilde{X}_{\text{CLS}} \\ \tilde{X}_1 \\ \dots \\ \tilde{X}_N \end{pmatrix} \quad , \tag{12}$$

where $\tilde{\mathbf{W}} = W_{\text{MLP}}^{(L)} \dots W_{\text{MLP}}^{(1)}$ and $\tilde{X}_i = (X_i \mathbf{E} + \mathbf{E}_{\text{pos,i}})\tilde{\mathbf{W}}$. With the normalization components in neural networks and the assumption of independence between the input tokens, $\{\tilde{X}_{\text{CLS}}, \tilde{X}_1 \dots, \tilde{X}_N\}$ forms a set of basis vectors.

**Remark.** The recurrence relation of Attention Rollout method (Abnar & Zuidema, 2020) can be rediscovered under the assumption that $W^{(l)} = I$. With this assumption, we have $\mathbf{A}^{(l)}\mathbf{Z}^{(l-1)}W^{(l)} + \mathbf{Z}^{(l-1)} = \mathbf{A}^{(l)}\mathbf{Z}^{(l-1)} + \mathbf{Z}^{(l-1)}$ so that $\tilde{\mathbf{A}}^{(l)} = \mathbf{A}^{(l)} + I$. Hence, we obtain the expression of Attention Rollout with the conclusions from Approximation 2: $\mathbf{Rollout} = (\mathbf{A}^{(L)} + I) \dots (\mathbf{A}^{(1)} + I)$.

However, the assumption of regrading it as an identity matrix is too strong. Although $W^{(l)}$ only operates on the embedding dimension, due to $\mathbf{A}^{(l)}$, it also contributes to a token interaction. In order to study its effect and find a more accurate $\tilde{\mathbf{A}}^{(l)}$ expression, we derive $\mathbf{O}^{(l)} = \mathbf{A}^{(l)}\mathbf{Z}^{(l-1)}W^{(l)}$ by two approximations: $\tilde{\mathbf{A}}_{\text{token}}^{(l)}$ and $\tilde{\mathbf{A}}_{\text{head}}^{(l)}$. We name the token-wise attention map using "token" since it contributes the effect of $W^{(l)}$ to $\mathbf{A}^{(l)}$ as a norm difference between token vectors, and name the head-wise one for a difference between heads.

**Token-wise Attention Map.** By the definition of matrix product, we develop $\mathbf{O} = \mathbf{A}\mathbf{Z}W$ and have

$$\mathbf{O}_{i,j} = \sum_m \mathbf{A}_{i,m}\left(\sum_k \frac{\mathbf{Z}_{m,k}W_{k,j}}{\mathbf{Z}_{m,j}}\right)\mathbf{Z}_{m,j} \overset{\text{token}}{\underset{\text{wise}}{\approx}} \sum_m \left(\mathbf{A}_{i,m}\frac{\|(\mathbf{Z}W)_{m,\cdot}\|}{\|\mathbf{Z}_{m,\cdot}\|}\right)\mathbf{Z}_{m,j} \tag{13}$$

$$\implies \quad \mathbf{O}^{(l)} = \mathbf{A}^{(l)}\mathbf{Z}^{(l-1)}W^{(l)} \overset{\text{token}}{\underset{\text{wise}}{\approx}} \tilde{\mathbf{A}}_{\text{token}}^{(l)}\mathbf{Z}^{(l-1)} \quad , \tag{14}$$

where $(\tilde{\mathbf{A}}_{\text{token}}^{(l)})_{\cdot,i} = \alpha_i \mathbf{A}_{\cdot,i}^{(l)}$ and $\alpha_i = \frac{\|(\mathbf{Z}^{(l-1)}W^{(l)})_{i,\cdot}\|}{\|\mathbf{Z}_{i,\cdot}^{(l-1)}\|}$. So with $\tilde{\mathbf{A}}_{\text{token}}^{(l)}$, we obtain $\mathbf{Z}^{(l)} \overset{\text{token}}{\underset{\text{wise}}{\approx}} (\tilde{\mathbf{A}}_{\text{token}}^{(l)} + I)\mathbf{Z}^{(l-1)}W_{\text{MLP}}^{(l)}$. More derivation details can be found in the appendix.

**Head-wise Attention Map.** We redevelop $\mathbf{O} = \mathbf{A}\mathbf{Z}W$ by taking different heads into consideration, and have

$$\mathbf{O}_{i,j} = \sum_h \sum_m \mathbf{A}_{i,m}^h\left(\sum_k \frac{\mathbf{Z}_{m,k}W_{k,j}^h}{\mathbf{Z}_{m,j}}\right)\mathbf{Z}_{m,j} \overset{\text{head}}{\underset{\text{wise}}{\approx}} \sum_h \theta_h(\mathbf{A}^h\mathbf{Z})_{i,j} \quad , \tag{15}$$

where $\mathbf{A}^h$ is the attention of head $h$ and $\theta_h$ can be regarded as a head weight. Recent works (Michel et al., 2019; Hao et al., 2021) propose a head importance concept $I_h$ by considering the sensitivity of head mask, which means that if $I_h$ is small, removing this head won't affect the model output a lot. A head importance expression is also provided in their work, with the inner product of the attention and its gradient: $I_h = (\mathbf{A}^h)^T \frac{\partial \mathcal{L}^c}{\partial \mathbf{A}^h}$. We adopt here the concept of $I_h$ with normalisation $\theta_h = \frac{I_h}{\sum_h I_h}$ to describe the head weight. Hence, we propose another head-wise type, which is easier to apply in practice, $\tilde{\mathbf{A}}_{\text{head}}^{(l)} = \sum_{h=1}^H \left(\frac{I_h^{(l)}}{\sum_h I_h^{(l)}}\right)\mathbf{A}_h^{(l)}$ and $\mathbf{Z}^{(l)} \overset{\text{head}}{\underset{\text{wise}}{\approx}} (\tilde{\mathbf{A}}_{\text{head}}^{(l)} + I)\mathbf{Z}^{(l-1)}W_{\text{MLP}}^{(l)}$.

With the above defined recurrence relations and Approximation 2, we obtain the tokens' interactions $\mathbf{P}^{(L)}$ until the last attention block $L$:

$$\mathbf{P}^{(L)} = (\tilde{\mathbf{A}}^{(L)} + I) \cdot \mathbf{P}^{(L-1)} = (\tilde{\mathbf{A}}^{(L)} + I) \dots (\tilde{\mathbf{A}}^{(1)} + I) \quad , \tag{16}$$

where $\tilde{\mathbf{A}}^{(l)}$ can be $\tilde{\mathbf{A}}_{\text{token}}^{(l)}$ from Eq. (14) or $\tilde{\mathbf{A}}_{\text{head}}^{(l)}$ from Eq. (15).

**Our token-wise attention perception achieves $20\times$ more accurate than Attention Rollout.** In order to learn the errors between different approximation strategies and the true value, we conduct several tests for Attention Rollout, token-wise and head-wise attention maps. In each block $l$, we compare $\mathbf{Z}_{\text{rollout}}^{(l)}$, $\mathbf{Z}_{\text{token}}^{(l)}$ and $\mathbf{Z}_{\text{head}}^{(l)}$ by $\mathbf{Z}_{\text{approx}}^{(l)} = (\tilde{\mathbf{A}}^{(l)} + I)\mathbf{Z}_{\text{true}}^{(l-1)}W_{\text{MLP}}^{(l)}$ with the true output $\mathbf{Z}_{\text{true}}^{(l)}$ and calculate the square error $\mathbf{E} = \|\mathbf{Z}_{\text{true}}^{(l)} - \mathbf{Z}_{\text{approx}}^{(l)}\|_2^2$ for each method.

Randomly selected 5000 images from ImageNet val set are used for the test. Both of the averages for our token-wise and head-wise methods are below Attention Rollout, especially the token-wise approximation achieves $20\times$ more accurate than Attention Rollout. Figure 3 presents the errors in different blocks, where we can see that our token-wise method has a more stable performance when the layer gets deeper.

### 3.3 Reasoning Feedback

Attention perception, taken for the first sub-process, is explained by $\mathbf{P}^{(L)}$ as introduced previously. It remains to the reasoning part $\mathbf{F}^c$, which refers to the stage of making the final decision based on the representations before the prediction head. We explain it by $\frac{\partial \mathcal{L}^c(X)}{\partial \mathbf{A}^{(L)}}$, the back-propagated gradients of the final prediction with respect to the attention map of the last block.

**Integrated Gradient for Token-level Importance.** Directly calculating the gradient of the last attention map may bring some noise and irrelevant features to the saliency map, due to the lack of sensitivity, which causes gradients to focus on the irrelevant features practically (Sundararajan et al., 2017; Smilkov et al., 2017). In order to reduce the gradient self-induced noise, we use an Integrated Gradient method (IG) (Sundararajan et al., 2017) to get the relevant gradients of the last attention map. We set the baseline to all zero, i.e. a black image or all padding tokens, and a linear path to the input. With Riemann approximation, we discretize the integration, having

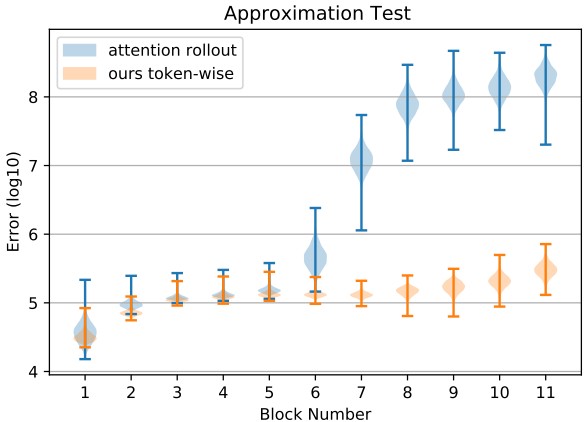

Figure 3: Results of our approximation tests. We notice that our token-wise approximation achieves 10∼100x more accurate than Attention Rollout in different blocks. Besides, the variance of errors in each block also decreases in our token-wise method.

$$\mathbf{F}^c = \text{ReLU}\left(\frac{1}{K}\sum_{k=1}^{K}\frac{\partial \mathcal{L}^c(\frac{k}{K}X)}{\partial \mathbf{A}^{(L)}}\right) \overset{K\to\infty}{\Rightarrow} \text{ReLU}\left(\int_{\alpha=0}^{1}\frac{\partial \mathcal{L}^c(\alpha X)}{\partial \mathbf{A}^{(L)}}d\alpha\right) \quad, \tag{17}$$

where only the positive part of the gradient is taken into consideration and K the total steps.

Combining the results of *Attention Perception* and *Reasoning Feedback*, our explanation method $\mathcal{T}_{\text{cls}}$ can be finally expressed as

$$\mathcal{T}_{\text{cls}} = \left(\mathbf{P}^{(L)}\odot\mathbf{F}^c\right)_{0,\{1...N\}} = \left(\mathbf{P}^{(L)}\odot\text{ReLU}\left(\frac{1}{K}\sum_{k=1}^{K}\frac{\partial \mathcal{L}^c(\frac{k}{K}X)}{\partial \mathbf{A}^{(L)}}\right)\right)_{0,\{1...N\}} \quad, \tag{18}$$

where $\mathbf{P}^{(L)} = (\tilde{\mathbf{A}}^{(1)} + I)(\tilde{\mathbf{A}}^{(2)} + I)...(\tilde{\mathbf{A}}^{(L)} + I)$ and two types of $\tilde{\mathbf{A}}^{(l)}$ are proposed: token-wise $\tilde{\mathbf{A}}^{(l)}_{\text{token}}$ in Eq. (14) and head-wise $\tilde{\mathbf{A}}^{(l)}_{\text{head}}$ in Eq. (15).

# 4 Experiments

We validate our proposed explanation method by comparing the results with several strong baselines. The experiment settings are based on two aspects: different modalities and different model versions. The experimental results show the clear advantages and wide applicability of our methods over the others in explaining Transformers.

## 4.1 Experimental Settings

**Faithfulness Evaluation.** Following previous works (Abnar & Zuidema, 2020; Chefer et al., 2021a;b; Samek et al., 2017; Vu et al., 2019; DeYoung et al., 2020), we prepare three types of tests for the trustworthiness evaluation:

1) **Perturbation Tests** gradually mask out the tokens of input according to the explanation results and measure the mean accuracy of all these masked inputs, i.e., area under the curve (AUC). There are two kinds of perturbation tests, positive and negative. In the positive perturbation test, tokens are masked from the highest (most relevant) to the lowest. A steep decrease (lower AUC) indicates that the relevance does reflect their importance to the classification score. And vice versa in the negative perturbation test. With the class-specificity for some methods, we adopt their explanation

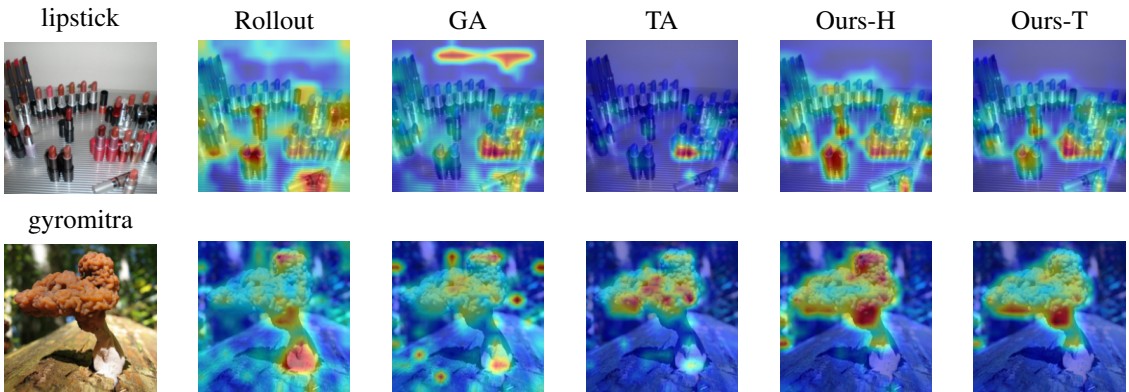

Figure 4: Localization of fine-grained regions for single class prediction of ViT-Base. Both samples are correctly predicted with a high probability score.

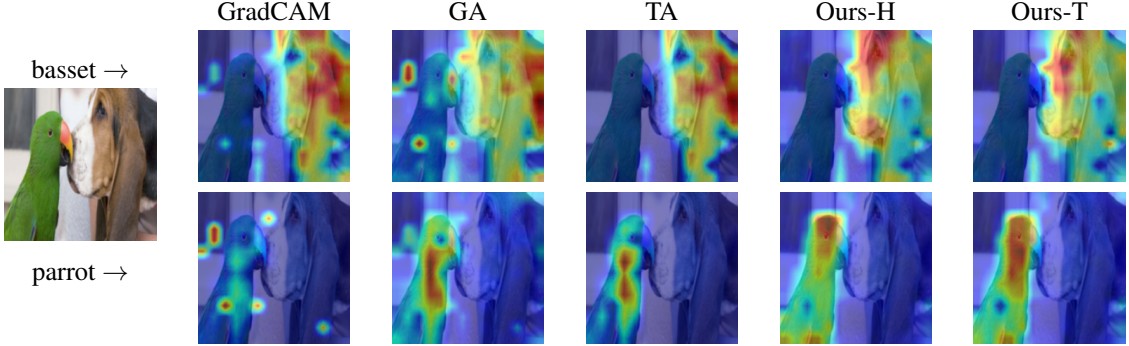

Figure 5: Class-specific visualization results of ViT-Base. We select the class discriminative methods at the very first place, and present the results of two different classes.

results on two classes: **Predicted**, which refers to the class with highest probability score, and **Target**, which refers to the target class.

2) **Segmentation Tests** equal the explanation to a semantic segmentation of image and compare it with the ground-truth in the dataset. We use here the predicted class to obtain the explanation results for class-specific methods. Each explanation can be evaluated by four metrics: pixel accuracy, mean Intersection over Union (mIoU), mean Average Precision (mAP) and mean F1 (mF1). Both pixel accuracy and mIoU are obtained after binarizing with a threshold set as the average of attribution scores. The metrics mAP and mF1 are calculated by averaging the corresponding scores at each threshold point.

3) **Language Reasoning** comes from a NLP benchmark ERASER (DeYoung et al., 2020) for rationales extraction, whose goal is to extract the input text spans that support the classification. Similar to the segmentation test, with the provided ground-truth, we measure the F1 score of the top-k tokens according to the relevance map, where we consider them to be part of the "rationale".

**Baselines and Ours.** The following six explanation methods are used as the baselines for a comparison in our experiments: Raw Attention Map (**RAM**), Attention Rollout (Abnar & Zuidema, 2020) (**Rollout**), Generic Attribution (Chefer et al., 2021b) (**GA**), GradCAM (Selvaraju et al., 2017) (**CAM**), Partial LRP (Voita et al., 2019) (**PLRP**) and Transformer Attribution (Chefer et al., 2021a) (**TA**). The appendix provides a detailed description of each method. We choose these baselines according to their explainability literature and applicability to the faithfulness evaluations. Black-box methods, which are computationally expensive and intrinsically different, are not considered here. As for our methods, we have, for each experiment, the token-wise (**Ours-T**) and the head-wise (**Ours-H**) types.

## 4.2 Transformers on Different Modalities

In this subsection, we focus on the explanation results of all methods with different modalities. We test them on three types of Transformers: **BERT** (Devlin et al., 2019) for texts, **ViT** (Dosovitskiy et al., 2021) for images and

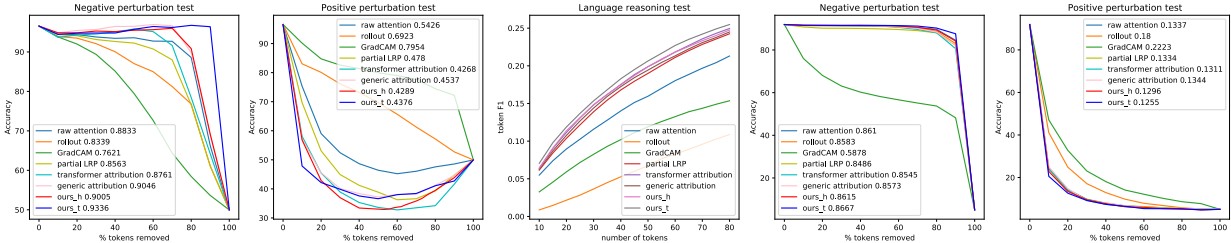

Figure 6: Results for BERT: *(from left to right)* Negative Perturbation Test on Movie Reviews, Positive Perturbation Test on Movie Review, Language Reasoning Test on Movie Reviews, Negative Perturbation Test on 20 Newsgroups and Positive Perturbation Test on 20 Newsgroups.

Table 1: Comparison of Positive (lower is better) and Negative (higher is better) perturbation AUC scores (percents) on ImageNet for ViT-Base, ViT-Large and ViT-MAE. H stands for the Head-Wise methods and T for the Token-Wise.

|  |  |  | RAM | Rollout | CAM | PLRP | GA | TA | Ours-H | Ours-T |
|---|---|---|---|---|---|---|---|---|---|---|
| **ViT-Base** | Positive | Predicted | 24.51 | 20.45 | 34.61 | 20.18 | 20.95 | 17.37 | 15.97 | **15.89** |
|  |  | Target | - | - | 34.07 | 20.20 | 20.01 | 16.65 | 15.14 | **15.13** |
|  | Negative | Predicted | 45.65 | 53.86 | 41.95 | 50.62 | 48.95 | 54.53 | 57.13 | **57.97** |
|  |  | Target | - | - | 42.46 | 50.64 | 51.23 | 55.70 | 58.54 | **59.24** |
| **ViT-Large** | Positive | Predicted | 27.71 | 21.91 | 45.66 | 30.19 | 19.21 | 20.41 | 18.01 | **17.99** |
|  |  | Target | - | - | 45.49 | 30.19 | 18.53 | 19.93 | 17.80 | **17.57** |
|  | Negative | Predicted | 40.99 | 53.44 | 47.58 | 37.14 | 54.72 | 52.67 | 55.86 | **56.44** |
|  |  | Target | - | - | 47.69 | 37.16 | 56.13 | 53.25 | 56.33 | **56.88** |
| **ViT-MAE** | Positive | Predicted | 38.56 | 38.20 | 56.79 | 26.59 | 33.11 | 34.00 | **20.57** | 20.72 |
|  |  | Target | - | - | 57.24 | 26.57 | 32.63 | 33.74 | **20.22** | 20.30 |
|  | Negative | Predicted | 40.79 | 52.03 | 24.80 | 55.34 | 57.67 | 56.92 | **65.33** | 64.48 |
|  |  | Target | - | - | 24.41 | 55.37 | 58.80 | 57.31 | **66.06** | 65.18 |

**CLIP** (Radford et al., 2021) for bi-modality. And we will present the experimental results by models. In terms of a Transformer literature and model size equivalence, we select BERT-base, ViT-base-16-224 and ViT-B/32 for CLIP, where all of them are derived from the Transformer Encoder architecture. The architectures of BERT and ViT are the same, except that BERT deals with (sub-)words while ViT copes with patches of an input image. CLIP adopts a dual encoder, in which text and image are respectively encoded and interact by cosine similarities.

**ViT.** Class-discriminability and the capacity of capturing fine-grained details are the criteria of a good visual explanation (Selvaraju et al., 2017). Figures 4 and 5 provide the visualizations of some images selected from ImageNet (Deng et al., 2009), demonstrating that our method is of such capacities. More samples can be found in appendix. As shown in the figures, Rollout captures a complete region of the target, but simply taking an average of attentions brings noises on the irrelevant pixels. TA and GA eliminate all the attentions with negative gradients which may result in an ignorance of some relevant pixels. Our methods perform well in capturing a very localized area without too much loss of the entire object, thanks to the combination of the attention perception that captures the entire object, and the reasoning feedback that decreases the noises of irrelevant pixels.

As for the quantitative evaluation, we select randomly 5 images per class (5k in total) from the ImageNet validation set for the perturbation tests, and the dataset of ImageNet-Segmentation (Guillaumin et al., 2014) for the segmentation tests. As shown in Tables 1 and 2, our methods outperform the other baselines on all metrics, especially the token-wise one, which leaves a great margin in segmentation tests, because of its more accurate approximation.

**BERT.** For the text modality, we first have the perturbation and language reasoning tests on the Movie Reviews Dataset (DeYoung et al., 2020) containing 1999 samples, which concerns a classification of positive and negative sentiments according to certain movie review. Our methods outperforms all the baselines in the language reasoning test.

Table 2: Segmentation test (percent, higher is better) on the ImageNet-Segmentation dataset for ViT-Base, ViT-Large and ViT-MAE.

|  |  | RAM | Rollout | CAM | PLRP | GA | TA | Ours-H | Ours-T |
|---|---|---|---|---|---|---|---|---|---|
| **ViT-Base** | mIoU | 46.37 | 55.42 | 41.30 | 57.95 | 58.34 | 61.92 | 60.74 | **66.32** |
|  | pixel accuracy | 67.87 | 73.54 | 65.91 | 76.31 | 76.30 | 79.68 | 78.04 | **82.15** |
|  | mAP | 80.24 | 84.76 | 71.60 | 84.67 | 85.28 | 85.99 | 86.18 | **88.04** |
|  | mF1 | 29.44 | 43.68 | 19.42 | 38.82 | 41.85 | 40.10 | 44.45 | **45.72** |
| **ViT-Large** | mIoU | 41.18 | 52.88 | 39.72 | 40.09 | 54.40 | 56.31 | 61.24 | **61.65** |
|  | pixel accuracy | 63.20 | 71.15 | 68.49 | 62.31 | 73.93 | 75.75 | **78.92** | 78.87 |
|  | mAP | 74.75 | 83.48 | 63.29 | 73.56 | 81.93 | 83.39 | 85.52 | **86.33** |
|  | mF1 | 25.58 | 42.76 | 10.25 | 23.99 | 38.04 | 38.18 | 42.40 | **43.38** |
| **ViT-MAE** | mIoU | 37.30 | 42.39 | 22.60 | 51.13 | 47.66 | 45.98 | **62.47** | 62.36 |
|  | pixel accuracy | 56.10 | 62.00 | 39.62 | 72.01 | 67.10 | 66.70 | **79.66** | 79.63 |
|  | mAP | 67.41 | 76.38 | 57.28 | 81.15 | 79.58 | 78.41 | **86.58** | 86.21 |
|  | mF1 | 31.99 | 34.20 | 12.99 | 33.94 | 36.51 | 32.43 | **44.67** | 44.08 |

For the perturbation test on Movie Reviews, the accuracy stops at a random guess (i.e. accuracy of 50%) at the end. We can see from Figure 6 that our methods outperform almost all the baselines, especially the token-wise method on the negative perturbation and language reasoning tests, where significant improvements are observed. As for the positive perturbation test, we see from the final score that TA is better than our methods, whereas in the first half of positive perturbation, the decrease according to both of our methods is clearly sharper than TA. We also notice that the methods with good performance on image, such as Rollout and GradCAM, lose its advantage for text. In contrast, our methods stay competitive for both text and image.

Since both of quantitative test for BERT are based on the Movie Review Dataset and the same fine-tuned classifier, we have another text perturbation test on 20 Newsgroups Dataset (Lang, 1995) for category classification. The 20 Newsgroups Dataset is a a collection of approximately 20,000 newsgroup documents across 20 different newsgroups. We finetune a BERT-base model on its training data, with the accuracy reaching 93% on testing set. We randomly select 3000 documents from the testing set for the perturbation. The results are consistent with the perturbation test on Movie Review, as shown in Figure 6, where GradCAM and Rollout lose their advantages and our methods still outperform the other baselines. We provide as well the visualizations for BERT in the appendix, where our methods show more human-understandable explanation results without too much focus on tokens like "the" or "and".

**CLIP.** A pretrained CLIP achieves good performances in some vision-language tasks, such as image captioning. Two visualization examples are shown in Figure 7. For images, the captured area of our method corresponds well with the text, even for multiple objects in a noun phrase. For texts, the captured tokens are also the key words in the noun phrase. More examples can be found in the appendix.

Besides, CLIP can also match the performance of original ResNet50 (He et al., 2016) on ImageNet for the zero-shot prediction task with a designed prompt, by calculating the similarities between "a photo of *class name*" and the corresponding image. For the perturbation test, we have a zero-shot image classification task with the same 5k images as ViT from the ImageNet validation set. As shown in Table 3, our methods outperform all the baselines as well.

## 4.3 Transformers of Different Versions

In this subsection, we test the applicability of all explanation algorithms to different versions of Transformers, for example, with different model depth and pooling strategies. Besides **ViT-Base** in Sub-section 4.2, we conduct evaluation experiments on another two variants: ViT-large-16-224 (Dosovitskiy et al., 2021), denoted **ViT-Large** and **ViT-MAE** with the global pooling strategy instead of [CLS] pooling in vanilla ViT-base-16-224 model. ViT-Large (ViT-L/16) is a very big model, which consist of 24 successive attention blocks and 16 attention heads with embedding size set as 1024. ViT-MAE adopts the same architecture as ViT-Base, but with a different pretraining approach and pooling strategy. The masked autoencoder (MAE) (He et al., 2022) is a self-supervised learning approach of masking random patches of the

Table 3: Comparison of Positive (lower is better) and Negative (higher is better) perturbation AUC scores (percents) on ImageNet for CLIP.

| | | RAM | Rollout | GradCAM | GA | Ours-H | Ours-T |
|---|---|---|---|---|---|---|---|
| Positive | Predicted | 14.11 | 19.32 | 21.49 | 12.78 | **12.29** | 12.32 |
| | Target | - | - | 21.34 | 11.91 | 11.55 | **11.48** |
| Negative | Predicted | 34.02 | 28.83 | 28.78 | 35.38 | **35.74** | 35.69 |
| | Target | - | - | 28.88 | 37.09 | **37.21** | 37.10 |

Figure 7: Visualization of bi-modal model: CLIP. The visualization results with our method correspond well with the highlighted input phrases, even for that with multiple objects.

input images and reconstructing the missing pixels. By reconstruction, the encoder learns more complex semantics, thus leading to better performances.

**ViT-Large.** We provide in the first line of Figure 8 the visualizations of ViT-Large for the class "tiger cat". It's interesting to see that the explanation methods, such as GradCAM, GA and TA, which capture the quite semantic tokens for ViT-base, focus on very irrelevant tokens for ViT-Large, whereas the ViT-Large shows better accuracy and transferability in the prediction. We believe that the irrationality results from an intuitive formulation of expression. With more successive attention blocks, the error accumulates, thus leading to less trustworthy visualizations. However, our methods, derived from explicit derivations, show far more understandable results in predicting the "tiger cat".

As for quantitative tests, we show in Table 2 and Table 1 the segmentation results and perturbation results for ViT-Large, where our methods outperform all the baselines in all the settings.

**ViT-MAE.** For all the baselines, their explanations of Transformer in global pooling setting are not provided in the original paper. To be more fair, we modify these methods by using the average of attribution scores from the second to the last line, indicating the first to the last token, instead of the first line for [CLS] token.

The visualization of ViT-MAE is shown in the third line of Figure 8 with the same photo and predicting class. Our methods not only capture the very semantic tokens, but also show a good coherence with the masked autoencoder approach, where by masking and reconstructing, the model can learn from more semantics. We credit the applicability of our methods in global pooling setting to the theoretical analysis as well.

Table 2 and Table 1 shows the results for segmentation and perturbation tests, where our methods leave a even more significant margin than the previous settings in both the positive and negative tests. It further demonstrates that our methods can also be applied to global pooled Transformers and provide trustworthy performances. Moreover, the

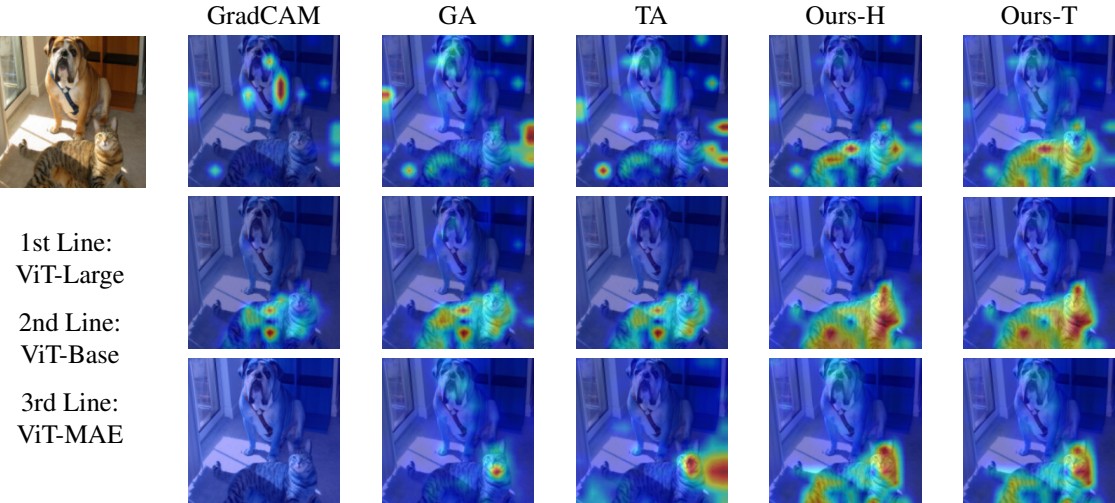

Figure 8: Visualizations of ViT-Large (1st line), ViT-Base (2nd line) and ViT-MAE (3rd line) on the same image for the same class *tiger cat*. Our methods show a more stable and less noisy capturing of semantic pixels than other baselines.

strategy of masking and reconstructing in MAE can alleviate the out-of-distribution issue in the perturbation test (Hase et al., 2021; Hooker et al., 2019) and make the quantitative results more convincing.

## 4.4 Ablation Study

We propose two ablation studies: the first focuses on the effectiveness of our expression, while the second focuses on the benefits of introducing the token-wise and head-wise coefficients. We conduct our ablation analyses with the same evaluation settings for ViT-Base, mentioned in Section 4.2. From Table 4, we notice that the results correspond well with our derivations in Section 3. Introducing token-wise and head-wise attention maps clearly enhances the quality of explanations. Moreover, the first ablation study also indicates the effectiveness of introducing the integrated version of gradients to reduce the noise.

Table 4: Results of ablation study with ViT-Base. Avg stands for a simple average of attentions for all heads and $\mathbf{A}^{(\mathbf{L})}$ for the attention map of the last block. $\mathbf{G}^{(\mathbf{L})}$ denotes the attention gradient of the last block and $\int \mathbf{G}^{(\mathbf{L})}$ the integrated version. We report the results of perturbation test with its predicted class. The first part is designed to analyze the effectiveness our derived attribution expression $\mathcal{T}$ and the second part for the befits of introducing our token-wise/head-wise attention maps.

| Setting | | Segmentation | | | | Perturbation | |
|---|---|---|---|---|---|---|---|
| Perception | Feedback | Acc. | mIoU | mAP | mF1 | Pos. | Neg. |
| $\times$ | $\int \mathbf{G}^{(\mathbf{L})}$ | 67.31 | 48.76 | 77.93 | 38.44 | 18.56 | 52.12 |
| Head-wise | $\times$ | 71.83 | 53.24 | 83.66 | 42.26 | 20.20 | 52.79 |
| Head-wise | $\mathbf{G}^{(\mathbf{L})}$ | 75.67 | 57.75 | 84.49 | 43.01 | 17.33 | 55.27 |
| Head-wise | $\int \mathbf{G}^{(\mathbf{L})}$ | **78.04** | **60.74** | **86.18** | **44.45** | **15.97** | **57.13** |
| $\mathbf{A}^{(L)}$ | $\int \mathbf{G}^{(\mathbf{L})}$ | 77.55 | 59.16 | 84.65 | 38.82 | 19.05 | 52.45 |
| Avg | $\int \mathbf{G}^{(\mathbf{L})}$ | 77.88 | 60.56 | 86.14 | 44.34 | 16.72 | 55.59 |
| Head-wise | $\int \mathbf{G}^{(\mathbf{L})}$ | 78.04 | 60.74 | 86.18 | 44.45 | 15.97 | 57.13 |
| Token-wise | $\int \mathbf{G}^{(\mathbf{L})}$ | **82.15** | **66.32** | **88.04** | **45.72** | **15.89** | **57.97** |

## 4.5 Token-wise/Head-wise

The token-wise and head-wise manners come from the different approximation approaches we use for the recurrence relation of each attention block, as discussed in Section 3. The token-wise one credits the effects of linear projection to the differences in the norms of token vectors while the head-wise one considers the head differences. As for

implementation, the token-wise method may involve extra modifications to obtain the intermediates, since they are not the direct output of any component in the neural network. The head-wise one has no such issue because it only demands the attention and attention gradients for calculation.

However, both visualizations and quantitative results demonstrate that the differences between these two methods are very limited , since they are both derived within the same framework. They both outperform all the baseline methods and are applicable to different types of Transformers, which contributes to the explicitness behind these two methods.

## 5  Conclusions

Most of the Transformer-based models give the prediction by first integrating different tokens via successive attention blocks, and then using the pooled vector of last token representations for a prediction task. In this work, we propose a novel explanation method based on gradients, the change of basis and the chain rule, dividing the entire process into attention perception and reasoning processes. Moreover, by taking into account the linear projections, we derive two attention maps (token-wise and head-wise) for the attention perception process, which are more accurate than traditional Attention Rollout. We show in the experiments that our method achieves a better performance than all baselines in a series of evaluation settings, on specifying the class information, capturing the fine-grained pixels and being applicable to different model sizes, pooling strategies and modalities.

Based on this tool, we can explore in depth those Transformer-based models in the fields of both computer vision and language processing, which gives a hand in debugging, training and evaluating different Transformers. In this work, we only deal with Transformer Encoder. Other model architectures, such as Encoder-Decoder (Yang et al., 2019) or Decoder types (Brown et al., 2020), will be tackled in our future work.

## Broader Impact Statement

Being explainable is an important criterion to implement the deep learning algorithms under real life settings. With the wide applications of Transformer-based models, our methods can be very helpful by bringing up a trustworthy explanation and increasing the transparency of model prediction. However, it may also be used for a malicious purpose if the user intends to trick the model with modifications to the most or the least important tokens.

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

# A  Table of Notations

Table 5: Table of notations

**Indices:**

| | |
|---|---|
| $L$ | the number of multi-head self-attention blocks ($l \in \{1, ..., L\}$), which is used as the superscript $(l)$ to denote the variables and parameters concerning the $l$-th block. |
| $N$ | the number of tokens/patches. |
| $C$ | the number of classes during the prediction ($c \in \{1, ..., C\}$). |
| $H$ | the number of heads in the self-attention block ($h \in \{1, ..., H\}$). |
| $K$ | the number of steps while discretizing the integration ($k \in \{1, ..., K\}$), which is only involved in Section 3.3. |
| $\mathcal{P}$ | the pooling strategy ($\mathcal{P} \in \{\mathrm{cls}, \mathrm{glp}, \dots\}$). |
| $i, j$ | the subscript indexing the matrix ($i \in \{0, 1, ..., N\}$ and 0 denotes the added [CLS] token). For example, $\mathbf{A}_{i,j}$ stands for the element at the $(i+1)$-th row and $(j+1)$-th column of the matrix $\mathbf{A}$. We also use $\mathbf{A}_{i,.}$ for the entire $(i+1)$-th row, or $\mathbf{A}_{0,\{1,2,\cdots,N\}}$ for the second to $N$-th columns at the first row. |

**Our Explanation Method:**

| | |
|---|---|
| $\mathcal{T}$ | expression of our explanation method |
| $\mathbf{P}$ | attention perception |
| $\mathbf{F}$ | reasoning feedback |
| $\tilde{\mathbf{A}}_{\mathrm{token}}$ | the token-wise attention map |
| $\alpha$ | the token-wise coefficients |
| $\tilde{\mathbf{A}}_{\mathrm{head}}$ | the head-wise attention map |
| $I$ | the head importance |
| $\theta$ | the head-wise coefficients |

**Transformer Parameters:**

| | |
|---|---|
| $\mathbf{E}$ | the embedding module |
| $\mathbf{E}_{\mathrm{pos}}$ | the positional embedding |
| $D$ | the embedding size |
| $D_{\mathrm{hidden}}$ | the hidden size |
| $W$ | the linear transformation matrix |
| $b$ | the bias in linear transformation |

**Variables:**

| | |
|---|---|
| $X$ | the input image patches or text tokens |
| $\mathbf{Z}$ | the input/output of self-attention block |
| $\tilde{\mathbf{Z}}$ | the output after multi-head self-attention module |
| $\mathbb{B}$ | the set of basis vectors representing each token |
| $\tilde{X}$ | the basis vector |
| $t$ | the coordinate of input vector over basis vector |
| $\mathbf{A}$ | the raw attention map |
| $Q$ | query |
| $K$ | key |
| $V$ | value |
| $\mathcal{L}$ | the loss function |

# B  Details of Baselines

**Raw attention map (RAM)** Raw attention map is a rather straightforward explanation method, which takes into consideration only the attention map of the last block. For the model with a [CLS] pooling method, it explains the

importance of different patches with the first row of the attention matrix. In our experiment, the last attention layer is visualized.

**Attention Rollout (Abnar & Zuidema, 2020) (Rollout)** Different from raw attention, all the attention maps along a forward-pass count for the explanation, where they assume a linear combination of tokens and quantify the influence of skip connections by adding an identity matrix. Note that both raw attention and Attention Rollout are class-agnostic.

**GradCAM (Selvaraju et al., 2017)** GradCAM provides a class-specific explanation, where they add weights to gradient based feature map of the last convolution layer for a ConvNet based model. Following the work of Chefer et al. (2021a), we use a weighted gradient map of the last attention block, which corresponds to the [CLS] token .

**Partial LRP (Voita et al., 2019) (PLRP)** Not only the attention map of each block is considered, the relevance propagation based methods also take the information flow inside the whole neural network into account. These methods are also class-agnostic, which means the visualization remains the same for the prediction of all classes. Here, we visualize the relevance map of the last attention layer.

**Transformer attribution (Chefer et al., 2021a) (TA)** Transformer attribution method is a state-of-the-art class-specific explanation method for Transformer. It combines relevancy and attention-map gradient by regarding the gradient as a weight to the relevance for certain prediction task.

**Generic attribution (Chefer et al., 2021b) (GA)** Generic attribution extends the usage of Transformer attribution to co-attention and self-attention based models, such as VisualBERT, LXMERT etc. and propose a more generic relevancy update rule. Meanwhile, it replaces the relevance of Transformer attribution with the attention layer in each block.

## C  Token-wise Attention Map

Here, we provide the mathematical derivation details of the Eq. (14) and Eq. (15). The goal is to develop the formula $\mathbf{O} = \mathbf{A}\mathbf{Z}W$, where $\mathbf{O} \in \mathbb{R}^{(N+1) \times D}$, $\mathbf{A} \in \mathbb{R}^{(N+1) \times (N+1)}$ and $W \in \mathbb{R}^{D \times D}$.

By the definition of matrix product, we have:

$$\mathbf{O}_{i,j} = \sum_{m=1}^{N+1} \sum_{k=1}^{D} \mathbf{A}_{i,m} \mathbf{Z}_{m,k} W_{k,j} \quad . \tag{19}$$

Since $\mathbf{A}_{i,m}$, $\mathbf{Z}_{m,k}$ and $W_{k,j} \in \mathbb{R}$, it is possible to write when $\mathbf{Z}_{m,j} \neq 0$ that,

$$\mathbf{O}_{i,j} = \sum_{m=1}^{N+1} \sum_{k=1}^{D} \mathbf{A}_{i,m} \frac{\mathbf{Z}_{m,k} W_{k,j}}{\mathbf{Z}_{m,j}} \mathbf{Z}_{m,j} = \sum_{m=1}^{N+1} (\sum_{k=1}^{D} \mathbf{A}_{i,m} \frac{\mathbf{Z}_{m,k} W_{k,j}}{\mathbf{Z}_{m,j}}) \mathbf{Z}_{m,j} \tag{20}$$

$$= \sum_{m=1}^{N+1} \frac{\mathbf{A}_{i,m}}{\mathbf{Z}_{m,j}} (\sum_{k=1}^{D} \mathbf{Z}_{m,k} W_{k,j}) \mathbf{Z}_{m,j} = \sum_{m=1}^{N+1} \frac{\mathbf{A}_{i,m}}{\mathbf{Z}_{m,j}} (\mathbf{Z}W)_{m,j} \mathbf{Z}_{m,j} \tag{21}$$

$$= \sum_{m=1}^{N+1} \mathbf{A}_{i,m} \frac{(\mathbf{Z}W)_{m,j}}{\mathbf{Z}_{m,j}} \mathbf{Z}_{m,j} \overset{\text{token}}{\underset{\text{wise}}{\approx}} \sum_{m=1}^{N+1} \mathbf{A}_{i,m} \frac{\|(\mathbf{Z}W)_{m,\cdot}\|}{\|\mathbf{Z}_{m,}\|} \mathbf{Z}_{m,j} \tag{22}$$

$$= \sum_{m=1}^{N+1} \tilde{\mathbf{A}}_{i,m} \mathbf{Z}_{m,j} = (\tilde{\mathbf{A}}_{\text{token}} \mathbf{Z})_{i,j} \quad . \tag{23}$$

Here comes the definition of our token-wise attention map.

## D  Discussion about Self-Supervised Learning

Self-supervised learning strategy can bring new properties to the model, especially to the Transformer-based models (Caron et al., 2021; Chen et al., 2021; He et al., 2022). DINO (Caron et al., 2021) is a self-supervised learning method

for Vision Transformer, with a specific design to reflect directly semantics via features. In such case, adopting simple explanation methods such as raw attention map might already be promising. In order to understand the differences that the self-supervised learning strategies can bring to explanation results, we add an additional segmentation test for ViT-Base-DINO on ImageNet-Segmentation (Guillaumin et al., 2014). We apply our token-wise/head-wise methods and compare them with raw attention map.

Table 6: Segmentation test on ImageNet-Segmentation dataset (percent) for ViT-DINO

|  |  | Raw Attention Map | Ours-H | Ours-T |
| --- | --- | --- | --- | --- |
|  | IoU | 65.79 | **67.42** | 66.42 |
| ViT-Base-DINO | Acc | 82.08 | **83.71** | 83.09 |
|  | mAP | 86.79 | **87.60** | 87.26 |

As shown in Table 6, self-supervised learning can bring differences in explaining certain models, whereas our methods still perform better on localizing semantics. We notice that the results of our methods for DINO is almost the same as Supervised ViT-Base in Table 2. This may indicate that there are some hidden and hard-to-get semantics in the supervised ViT-Base that are equal to ViT-DINO. And our methods succeed in revealing this information and being applicable to all Transformer-based models.

Moreover, the ImageNet-Segmentation dataset only contains images with single objects. For explanation methods that are not class-specific, the segmentation results can be ambiguous when it appears multiple objects on the image. We show in Figure 9 the visualizations for multiple objects, which further validate the benefits of our methods in capturing semantics.

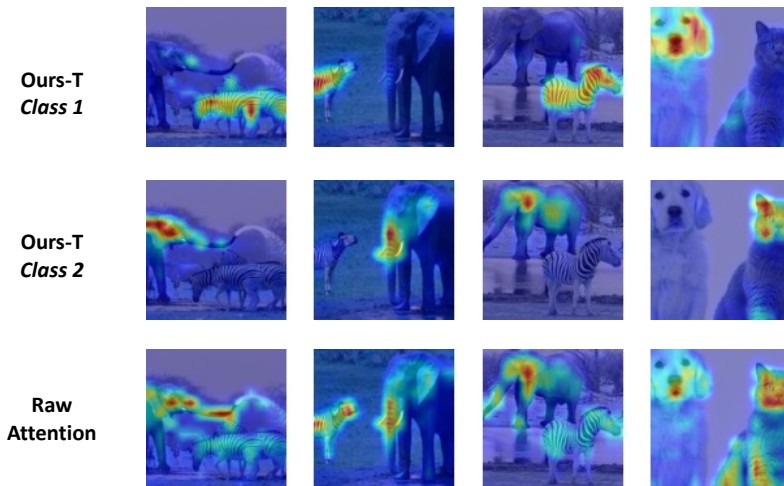

Figure 9: Class-specific visualizations for ViT-Base-DINO.

## E   More Samples, BERT and ViTs (Supervised, MoCo-v3, DINO and MAE)

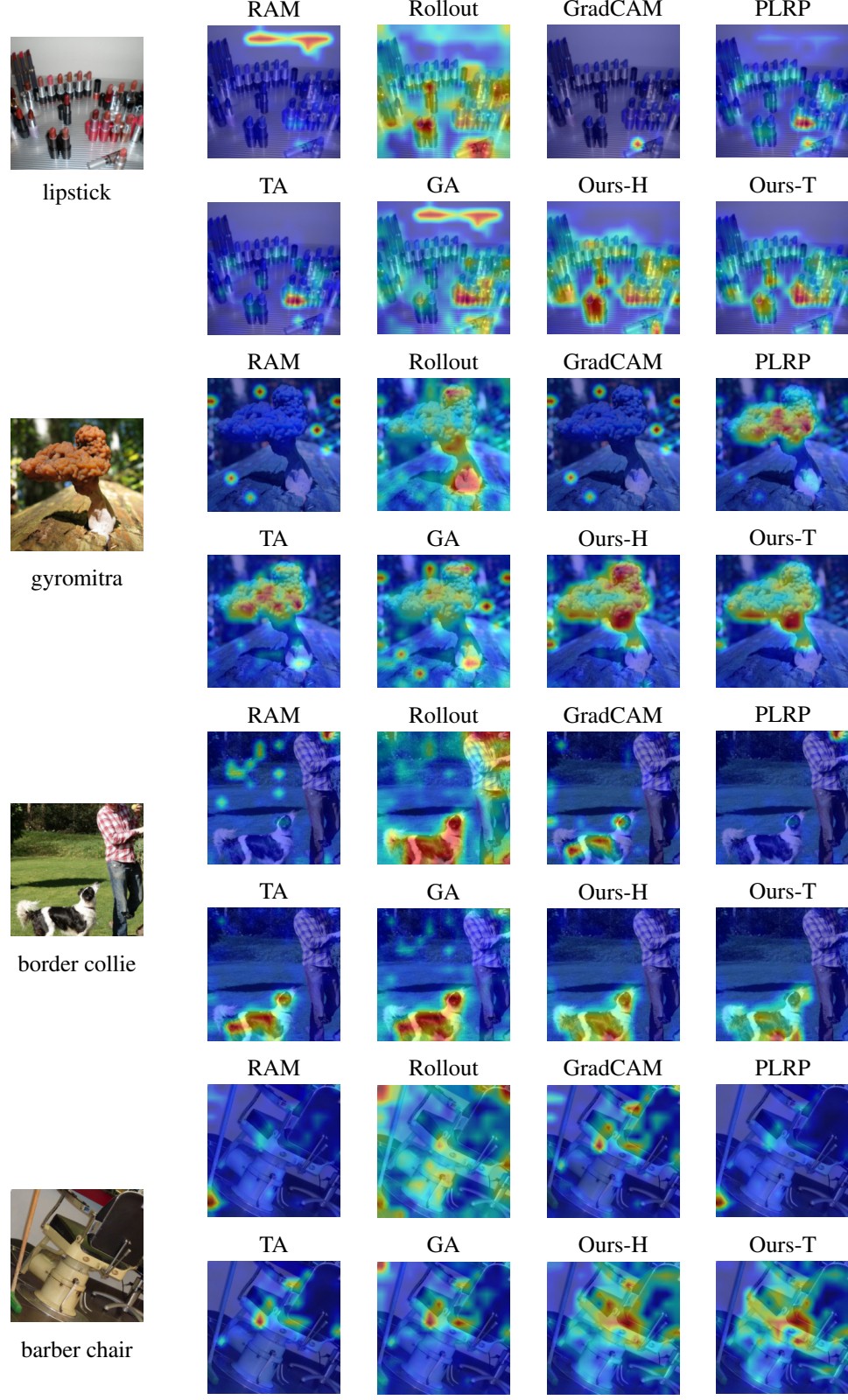

Figure 10: More samples of localization of fine-grained regions for single class prediction of ViT with full baselines.

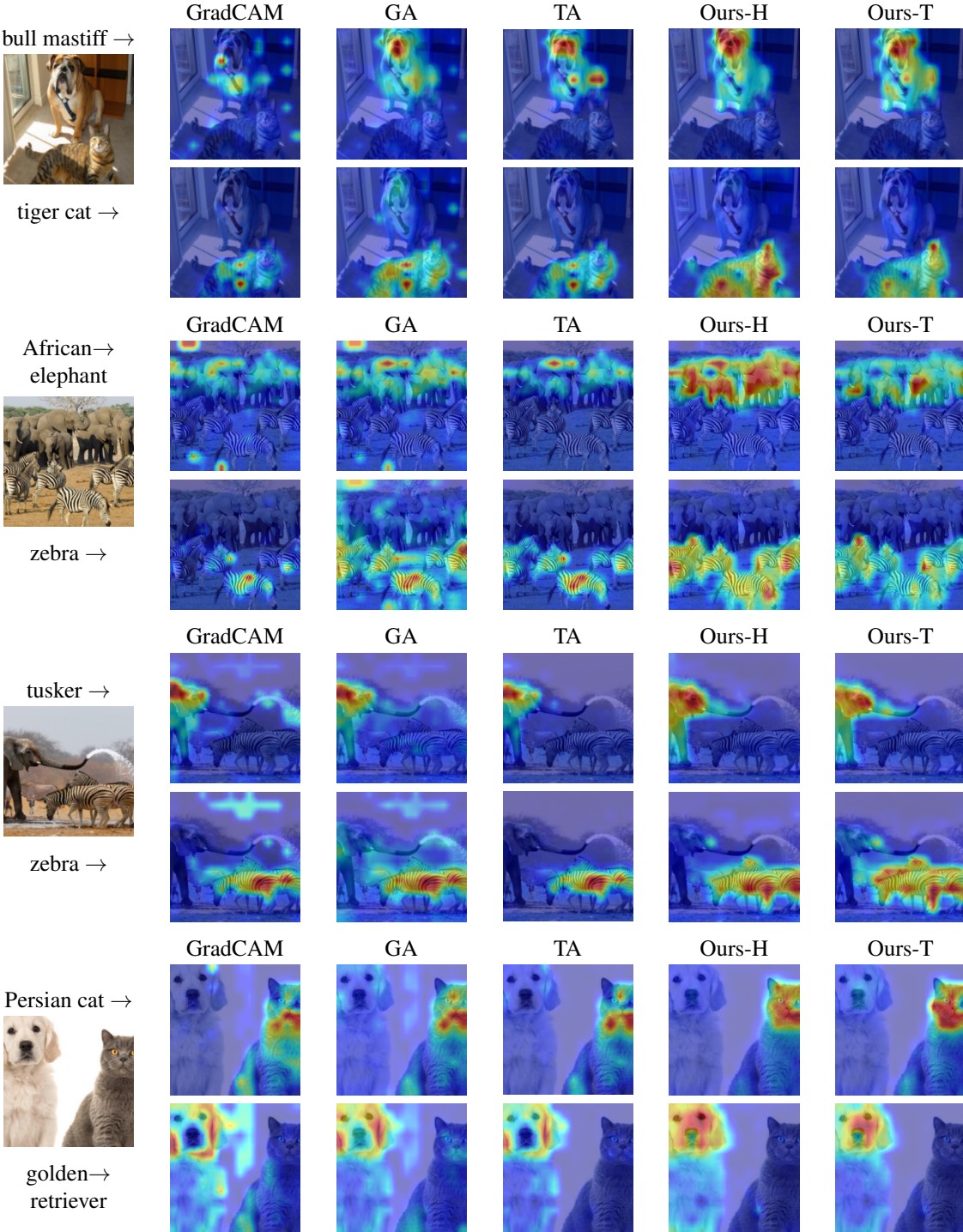

Figure 11: More samples of class-specific visualization results of ViT. We just present the results of two classes with class-specific baseline methods, while for the other methods, there are not any differences between different classes.

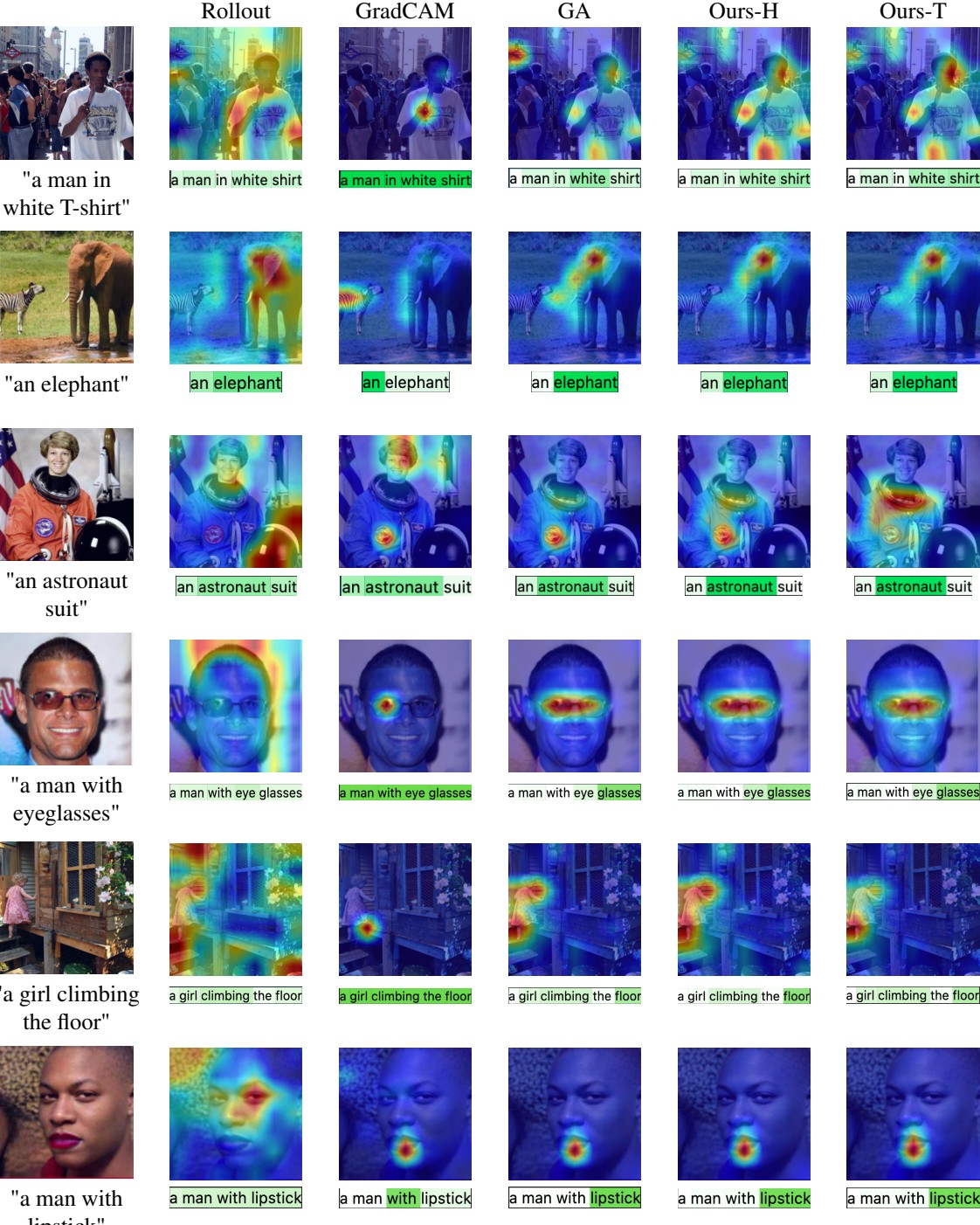

Figure 12: More samples of CLIP visualization. Both text and image visualizations are provided here.

## RAM

[CLS] ah , and 1999 was going along so well , too . " she ' s all that " has the dubious distinction of being the worst movie i ' ve seen so far this year . and quite frankly , i doubt i ' ll see anything equally bad . ( at least , i * hope * i do n ' t see anything equally bad ) . " she ' s all that " tells the story of the most popular guy in school ( played by freddie prinze jr . ) who accepts a bet to transform the geekiest girl in school ( rachel leigh cook ) into the most popular . that , right there , is problem # 1 . how many times have we seen this storyline ? as cook comments near the end of the film , " it ' s kind of like " pretty woman " , except without the prostitution " . of course , had the filmmakers attempted to try something new with this material , the well - worn storyline would have been a device to propel the movie forward . as it is , though , " she ' s all that " relies * completely * on the lame and overused formula to push it ahead . there ' s not one original or interesting character in the film , either , and if that was n ' t bad enough , there ' s not one good performance featured . the star of the movie , rachel leigh cook , is simply horrible . i usually do n ' t like to get so personal , but in this case , i think it needs to be said . cook wears the same expression throughout the flick and looks to be having as miserable a time as i was . i was never convinced that she was a " nerd " , and her transformation was unconvincing and unnecessary . the movie seems to be saying it ' s better to be popular than to be who you are . as for freddie prinze jr . , an actor i ordinarily enjoy , he too is quite bad here . he coasts through the film on so - called charm , and never establishes a real character . kieren culkin is here , too , as the brother of cook . and for some indiscernable reason , he ' s got hearing aids . no explanation is given and they ' re never brought up . were we supposed to feel * sorry * for him just because he wore hearing aids ? i do n ' t think so . that single element of the film was one of the most offensive things i ' ve seen in a movie in a long time . " she ' s all that " sucks [SEP]

## Rollout

[CLS] ah , and 1999 was going along so well , too . " she ' s all that " has the dubious distinction of being the worst movie i ' ve seen so far this year . and quite frankly , i doubt i ' ll see anything equally bad . ( at least , i * hope * i do n ' t see anything equally bad ) . " she ' s all that " tells the story of the most popular guy in school ( played by freddie prinze jr . ) who accepts a bet to transform the geekiest girl in school ( rachel leigh cook ) into the most popular . that , right there , is problem # 1 . how many times have we seen this storyline ? as cook comments near the end of the film , " it ' s kind of like " pretty woman " , except without the prostitution " . of course , had the filmmakers attempted to try something new with this material , the well - worn storyline would have been a device to propel the movie forward . as it is , though , " she ' s all that " relies * completely * on the lame and overused formula to push it ahead . there ' s not one original or interesting character in the film , either , and if that was n ' t bad enough , there ' s not one good performance featured . the star of the movie , rachel leigh cook , is simply horrible . i usually do n ' t like to get so personal , but in this case , i think it needs to be said . cook wears the same expression throughout the flick and looks to be having as miserable a time as i was . i was never convinced that she was a " nerd " , and her transformation was unconvincing and unnecessary . the movie seems to be saying it ' s better to be popular than to be who you are . as for freddie prinze jr . , an actor i ordinarily enjoy , he too is quite bad here . he coasts through the film on so - called charm , and never establishes a real character . kieren culkin is here , too , as the brother of cook . and for some indiscernable reason , he ' s got hearing aids . no explanation is given and they ' re never brought up . were we supposed to feel * sorry * for him just because he wore hearing aids ? i do n ' t think so . that single element of the film was one of the most offensive things i ' ve seen in a movie in a long time . " she ' s all that " sucks [SEP]

## GradCAM

[CLS] ah , and 1999 was going along so well , too . " she ' s all that " has the dubious distinction of being the worst movie i ' ve seen so far this year . and quite frankly , i doubt i ' ll see anything equally bad . ( at least , i * hope * i do n ' t see anything equally bad ) . " she ' s all that " tells the story of the most popular guy in school ( played by freddie prinze jr . ) who accepts a bet to transform the geekiest girl in school ( rachel leigh cook ) into the most popular . that , right there , is problem # 1 . how many times have we seen this storyline ? as cook comments near the end of the film , " it ' s kind of like " pretty woman " , except without the prostitution " . of course , had the filmmakers attempted to try something new with this material , the well - worn storyline would have been a device to propel the movie forward . as it is , though , " she ' s all that " relies * completely * on the lame and overused formula to push it ahead . there ' s not one original or interesting character in the film , either , and if that was n ' t bad enough , there ' s not one good performance featured . the star of the movie , rachel leigh cook , is simply horrible . i usually do n ' t like to get so personal , but in this case , i think it needs to be said . cook wears the same expression throughout the flick and looks to be having as miserable a time as i was . i was never convinced that she was a " nerd " , and her transformation was unconvincing and unnecessary . the movie seems to be saying it ' s better to be popular than to be who you are . as for freddie prinze jr . , an actor i ordinarily enjoy , he too is quite bad here . he coasts through the film on so - called charm , and never establishes a real character . kieren culkin is here , too , as the brother of cook . and for some indiscernable reason , he ' s got hearing aids . no explanation is given and they ' re never brought up . were we supposed to feel * sorry * for him just because he wore hearing aids ? i do n ' t think so . that single element of the film was one of the most offensive things i ' ve seen in a movie in a long time . " she ' s all that " sucks [SEP]

## PLRP

[CLS] ah , and 1999 was going along so well , too . " she ' s all that " has the dubious distinction of being the worst movie i ' ve seen so far this year . and quite frankly , i doubt i ' ll see anything equally bad . ( at least , i * hope * i do n ' t see anything equally bad ) . " she ' s all that " tells the story of the most popular guy in school ( played by freddie prinze jr . ) who accepts a bet to transform the geekiest girl in school ( rachel leigh cook ) into the most popular . that , right there , is problem # 1 . how many times have we seen this storyline ? as cook comments near the end of the film , " it ' s kind of like " pretty woman " , except without the prostitution " . of course , had the filmmakers attempted to try something new with this material , the well - worn storyline would have been a device to propel the movie forward . as it is , though , " she ' s all that " relies * completely * on the lame and overused formula to push it ahead . there ' s not one original or interesting character in the film , either , and if that was n ' t bad enough , there ' s not one good performance featured . the star of the movie , rachel leigh cook , is simply horrible . i usually do n ' t like to get so personal , but in this case , i think it needs to be said . cook wears the same expression throughout the flick and looks to be having as miserable a time as i was . i was never convinced that she was a " nerd " , and her transformation was unconvincing and unnecessary . the movie seems to be saying it ' s better to be popular than to be who you are . as for freddie prinze jr . , an actor i ordinarily enjoy , he too is quite bad here . he coasts through the film on so - called charm , and never establishes a real character . kieren culkin is here , too , as the brother of cook . and for some indiscernable reason , he ' s got hearing aids . no explanation is given and they ' re never brought up . were we supposed to feel * sorry * for him just because he wore hearing aids ? i do n ' t think so . that single element of the film was one of the most offensive things i ' ve seen in a movie in a long time . " she ' s all that " sucks [SEP]

## TA

[CLS] ah , and 1999 was going along so well , too . " she ' s all that " has the dubious distinction of being the worst movie i ' ve seen so far this year . and quite frankly , i doubt i ' ll see anything equally bad . ( at least , i * hope * i do n ' t see anything equally bad ) . " she ' s all that " tells the story of the most popular guy in school ( played by freddie prinze jr . ) who accepts a bet to transform the geekiest girl in school ( rachel leigh cook ) into the most popular . that , right there , is problem # 1 . how many times have we seen this storyline ? as cook comments near the end of the film , " it ' s kind of like " pretty woman " , except without the prostitution " . of course , had the filmmakers attempted to try something new with this material , the well - worn storyline would have been a device to propel the movie forward . as it is , though , " she ' s all that " relies * completely * on the lame and overused formula to push it ahead . there ' s not one original or interesting character in the film , either , and if that was n ' t bad enough , there ' s not one good performance featured . the star of the movie , rachel leigh cook , is simply horrible . i usually do n ' t like to get so personal , but in this case , i think it needs to be said . cook wears the same expression throughout the flick and looks to be having as miserable a time as i was . i was never convinced that she was a " nerd " , and her transformation was unconvincing and unnecessary . the movie seems to be saying it ' s better to be popular than to be who you are . as for freddie prinze jr . , an actor i ordinarily enjoy , he too is quite bad here . he coasts through the film on so - called charm , and never establishes a real character . kieren culkin is here , too , as the brother of cook . and for some indiscernable reason , he ' s got hearing aids . no explanation is given and they ' re never brought up . were we supposed to feel * sorry * for him just because he wore hearing aids ? i do n ' t think so . that single element of the film was one of the most offensive things i ' ve seen in a movie in a long time . " she ' s all that " sucks [SEP]

## GA

[CLS] ah , and 1999 was going along so well , too . " she ' s all that " has the dubious distinction of being the worst movie i ' ve seen so far this year . and quite frankly , i doubt i ' ll see anything equally bad . ( at least , i * hope * i do n ' t see anything equally bad ) . " she ' s all that " tells the story of the most popular guy in school ( played by freddie prinze jr . ) who accepts a bet to transform the geekiest girl in school ( rachel leigh cook ) into the most popular . that , right there , is problem # 1 . how many times have we seen this storyline ? as cook comments near the end of the film , " it ' s kind of like " pretty woman " , except without the prostitution " . of course , had the filmmakers attempted to try something new with this material , the well - worn storyline would have been a device to propel the movie forward . as it is , though , " she ' s all that " relies * completely * on the lame and overused formula to push it ahead . there ' s not one original or interesting character in the film , either , and if that was n ' t bad enough , there ' s not one good performance featured . the star of the movie , rachel leigh cook , is simply horrible . i usually do n ' t like to get so personal , but in this case , i think it needs to be said . cook wears the same expression throughout the flick and looks to be having as miserable a time as i was . i was never convinced that she was a " nerd " , and her transformation was unconvincing and unnecessary . the movie seems to be saying it ' s better to be popular than to be who you are . as for freddie prinze jr . , an actor i ordinarily enjoy , he too is quite bad here . he coasts through the film on so - called charm , and never establishes a real character . kieren culkin is here , too , as the brother of cook . and for some indiscernable reason , he ' s got hearing aids . no explanation is given and they ' re never brought up . were we supposed to feel * sorry * for him just because he wore hearing aids ? i do n ' t think so . that single element of the film was one of the most offensive things i ' ve seen in a movie in a long time . " she ' s all that " sucks [SEP]

## Ours-H

[CLS] ah , and 1999 was going along so well , too . " she ' s all that " has the dubious distinction of being the worst movie i ' ve seen so far this year . and quite frankly , i doubt i ' ll see anything equally bad . ( at least , i * hope * i do n ' t see anything equally bad ) . " she ' s all that " tells the story of the most popular guy in school ( played by freddie prinze jr . ) who accepts a bet to transform the geekiest girl in school ( rachel leigh cook ) into the most popular . that , right there , is problem # 1 . how many times have we seen this storyline ? as cook comments near the end of the film , " it ' s kind of like " pretty woman " , except without the prostitution " . of course , had the filmmakers attempted to try something new with this material , the well - worn storyline would have been a device to propel the movie forward . as it is , though , " she ' s all that " relies * completely * on the lame and overused formula to push it ahead . there ' s not one original or interesting character in the film , either , and if that was n ' t bad enough , there ' s not one good performance featured . the star of the movie , rachel leigh cook , is simply horrible . i usually do n ' t like to get so personal , but in this case , i think it needs to be said . cook wears the same expression throughout the flick and looks to be having as miserable a time as i was . i was never convinced that she was a " nerd " , and her transformation was unconvincing and unnecessary . the movie seems to be saying it ' s better to be popular than to be who you are . as for freddie prinze jr . , an actor i ordinarily enjoy , he too is quite bad here . he coasts through the film on so - called charm , and never establishes a real character . kieren culkin is here , too , as the brother of cook . and for some indiscernable reason , he ' s got hearing aids . no explanation is given and they ' re never brought up . were we supposed to feel * sorry * for him just because he wore hearing aids ? i do n ' t think so . that single element of the film was one of the most offensive things i ' ve seen in a movie in a long time . " she ' s all that " sucks [SEP]

## Ours-T

[CLS] ah , and 1999 was going along so well , too . " she ' s all that " has the dubious distinction of being the worst movie i ' ve seen so far this year . and quite frankly , i doubt i ' ll see anything equally bad . ( at least , i * hope * i do n ' t see anything equally bad ) . " she ' s all that " tells the story of the most popular guy in school ( played by freddie prinze jr . ) who accepts a bet to transform the geekiest girl in school ( rachel leigh cook ) into the most popular . that , right there , is problem # 1 . how many times have we seen this storyline ? as cook comments near the end of the film , " it ' s kind of like " pretty woman " , except without the prostitution " . of course , had the filmmakers attempted to try something new with this material , the well - worn storyline would have been a device to propel the movie forward . as it is , though , " she ' s all that " relies * completely * on the lame and overused formula to push it ahead . there ' s not one original or interesting character in the film , either , and if that was n ' t bad enough , there ' s not one good performance featured . the star of the movie , rachel leigh cook , is simply horrible . i usually do n ' t like to get so personal , but in this case , i think it needs to be said . cook wears the same expression throughout the flick and looks to be having as miserable a time as i was . i was never convinced that she was a " nerd " , and her transformation was unconvincing and unnecessary . the movie seems to be saying it ' s better to be popular than to be who you are . as for freddie prinze jr . , an actor i ordinarily enjoy , he too is quite bad here . he coasts through the film on so - called charm , and never establishes a real character . kieren culkin is here , too , as the brother of cook . and for some indiscernable reason , he ' s got hearing aids . no explanation is given and they ' re never brought up . were we supposed to feel * sorry * for him just because he wore hearing aids ? i do n ' t think so . that single element of the film was one of the most offensive things i ' ve seen in a movie in a long time . " she ' s all that " sucks [SEP]

Figure 13: BERT visualizations for a sample with negative emotion in Movie Reviews Dataset. A darker color represents a greater token attribution during the prediction.

### RAM

[CLS] in this good natured , pleasent and easy going comedy , bill murray ( ghostbusters , 1984 ) plays grumpy weatherman phil conners , who , every year , is sent to punxsutawney , p . a . to report on groundhog day . the groundhog day ceremony involves a groundhog being lifted out of a box , and if he dosen ' t see a shadow , it will be an early spring . phil really hates the ceremony , and not even his producer rita ( macdowell ) can change his mind . however , fate has a cruel trick for phil , and he starts re - living groundhog day over , and over , and over , until he gradually likes it , and rita falls in love with him . groundhog day is a well written , totally unoffensive and funny comedy . the screenwriters , director ramis and danny rubin , have written a funny , warm , but never overly senitmental comedy . although the idea of a day repeating over and over may sound tedious , there are enough good jokes to hold the audiences attention throughout the whole film . ramis ' s direction also helps , and although he dosen ' t try any flash director tricks , the film is directed well enough , and the jokes are set up well . and the editing is also good , especially when it shows one part of the day over , and over again , such as when phil tries to have the perfect night with rita . the performances are also excellent . bill murray is great fun , and his transistion from cynical to happy is smooth , and delivers his lines in his usual smary style . andie macdowell is good as rita , although sometimes she is just a little bit too sweet in some parts of the film . there is chemistry between the two leads , thankfully , otherwise the whole film would probably fall part . chris elliot , as the cameraman larry , is also funny , although you have to like his goofy style , otherwise you are really going to hate him throughout this film , and it will lower your enjoyment of the film overall . the supporting cast are n ' t bad either , with stephen tobolowsky hillarous as phils old school mate ned ryanson , and even director ramis popping up as a neurologist . in fact , there is not one dud performance in this film , and even the groundhog gets a funny scene involving a car chase . overall , there is really nothing wrong with groundhog day at all . [SEP]

### Rollout

[CLS] in this good natured , pleasent and easy going comedy , bill murray ( ghostbusters , 1984 ) plays grumpy weatherman phil conners , who , every year , is sent to punxsutawney , p . a . to report on groundhog day . the groundhog day ceremony involves a groundhog being lifted out of a box , and if he dosen ' t see a shadow , it will be an early spring . phil really hates the ceremony , and not even his producer rita ( macdowell ) can change his mind . however , fate has a cruel trick for phil , and he starts re - living groundhog day over , and over , and over , until he gradually likes it , and rita falls in love with him . groundhog day is a well written , totally unoffensive and funny comedy . the screenwriters , director ramis and danny rubin , have written a funny , warm , but never overly senitmental comedy . although the idea of a day repeating over and over may sound tedious , there are enough good jokes to hold the audiences attention throughout the whole film . ramis ' s direction also helps , and although he dosen ' t try any flash director tricks , the film is directed well enough , and the jokes are set up well . and the editing is also good , especially when it shows one part of the day over , and over again , such as when phil tries to have the perfect night with rita . the performances are also excellent . bill murray is great fun , and his transistion from cynical to happy is smooth , and delivers his lines in his usual smary style . andie macdowell is good as rita , although sometimes she is just a little bit too sweet in some parts of the film . there is chemistry between the two leads , thankfully , otherwise the whole film would probably fall part . chris elliot , as the cameraman larry , is also funny , although you have to like his goofy style , otherwise you are really going to hate him throughout this film , and it will lower your enjoyment of the film overall . the supporting cast are n ' t bad either , with stephen tobolowsky hillarous as phils old school mate ned ryanson , and even director ramis popping up as a neurologist . in fact , there is not one dud performance in this film , and even the groundhog gets a funny scene involving a car chase . overall , there is really nothing wrong with groundhog day at all . [SEP]

### GradCAM

[CLS] in this good natured , pleasent and easy going comedy , bill murray ( ghostbusters , 1984 ) plays grumpy weatherman phil conners , who , every year , is sent to punxsutawney , p . a . to report on groundhog day . the groundhog day ceremony involves a groundhog being lifted out of a box , and if he dosen ' t see a shadow , it will be an early spring . phil really hates the ceremony , and not even his producer rita ( macdowell ) can change his mind . however , fate has a cruel trick for phil , and he starts re - living groundhog day over , and over , and over , until he gradually likes it , and rita falls in love with him . groundhog day is a well written , totally unoffensive and funny comedy . the screenwriters , director ramis and danny rubin , have written a funny , warm , but never overly senitmental comedy . although the idea of a day repeating over and over may sound tedious , there are enough good jokes to hold the audiences attention throughout the whole film . ramis ' s direction also helps , and although he dosen ' t try any flash director tricks , the film is directed well enough , and the jokes are set up well . and the editing is also good , especially when it shows one part of the day over , and over again , such as when phil tries to have the perfect night with rita . the performances are also excellent . bill murray is great fun , and his transistion from cynical to happy is smooth , and delivers his lines in his usual smary style . andie macdowell is good as rita , although sometimes she is just a little bit too sweet in some parts of the film . there is chemistry between the two leads , thankfully , otherwise the whole film would probably fall part . chris elliot , as the cameraman larry , is also funny , although you have to like his goofy style , otherwise you are really going to hate him throughout this film , and it will lower your enjoyment of the film overall . the supporting cast are n ' t bad either , with stephen tobolowsky hillarous as phils old school mate ned ryanson , and even director ramis popping up as a neurologist . in fact , there is not one dud performance in this film , and even the groundhog gets a funny scene involving a car chase . overall , there is really nothing wrong with groundhog day at all . [SEP]

### PLRP

[CLS] in this good natured , pleasent and easy going comedy , bill murray ( ghostbusters , 1984 ) plays grumpy weatherman phil conners , who , every year , is sent to punxsutawney , p . a . to report on groundhog day . the groundhog day ceremony involves a groundhog being lifted out of a box , and if he dosen ' t see a shadow , it will be an early spring . phil really hates the ceremony , and not even his producer rita ( macdowell ) can change his mind . however , fate has a cruel trick for phil , and he starts re - living groundhog day over , and over , and over , until he gradually likes it , and rita falls in love with him . groundhog day is a well written , totally unoffensive and funny comedy . the screenwriters , director ramis and danny rubin , have written a funny , warm , but never overly senitmental comedy . although the idea of a day repeating over and over may sound tedious , there are enough good jokes to hold the audiences attention throughout the whole film . ramis ' s direction also helps , and although he dosen ' t try any flash director tricks , the film is directed well enough , and the jokes are set up well . and the editing is also good , especially when it shows one part of the day over , and over again , such as when phil tries to have the perfect night with rita . the performances are also excellent . bill murray is great fun , and his transistion from cynical to happy is smooth , and delivers his lines in his usual smary style . andie macdowell is good as rita , although sometimes she is just a little bit too sweet in some parts of the film . there is chemistry between the two leads , thankfully , otherwise the whole film would probably fall part . chris elliot , as the cameraman larry , is also funny , although you have to like his goofy style , otherwise you are really going to hate him throughout this film , and it will lower your enjoyment of the film overall . the supporting cast are n ' t bad either , with stephen tobolowsky hillarous as phils old school mate ned ryanson , and even director ramis popping up as a neurologist . in fact , there is not one dud performance in this film , and even the groundhog gets a funny scene involving a car chase . overall , there is really nothing wrong with groundhog day at all . [SEP]

### TA

[CLS] in this good natured , pleasent and easy going comedy , bill murray ( ghostbusters , 1984 ) plays grumpy weatherman phil conners , who , every year , is sent to punxsutawney , p . a . to report on groundhog day . the groundhog day ceremony involves a groundhog being lifted out of a box , and if he dosen ' t see a shadow , it will be an early spring . phil really hates the ceremony , and not even his producer rita ( macdowell ) can change his mind . however , fate has a cruel trick for phil , and he starts re - living groundhog day over , and over , and over , until he gradually likes it , and rita falls in love with him . groundhog day is a well written , totally unoffensive and funny comedy . the screenwriters , director ramis and danny rubin , have written a funny , warm , but never overly senitmental comedy . although the idea of a day repeating over and over may sound tedious , there are enough good jokes to hold the audiences attention throughout the whole film . ramis ' s direction also helps , and although he dosen ' t try any flash director tricks , the film is directed well enough , and the jokes are set up well . and the editing is also good , especially when it shows one part of the day over , and over again , such as when phil tries to have the perfect night with rita . the performances are also excellent . bill murray is great fun , and his transistion from cynical to happy is smooth , and delivers his lines in his usual smary style . andie macdowell is good as rita , although sometimes she is just a little bit too sweet in some parts of the film . there is chemistry between the two leads , thankfully , otherwise the whole film would probably fall part . chris elliot , as the cameraman larry , is also funny , although you have to like his goofy style , otherwise you are really going to hate him throughout this film , and it will lower your enjoyment of the film overall . the supporting cast are n ' t bad either , with stephen tobolowsky hillarous as phils old school mate ned ryanson , and even director ramis popping up as a neurologist . in fact , there is not one dud performance in this film , and even the groundhog gets a funny scene involving a car chase . overall , there is really nothing wrong with groundhog day at all . [SEP]

### GA

[CLS] in this good natured , pleasent and easy going comedy , bill murray ( ghostbusters , 1984 ) plays grumpy weatherman phil conners , who , every year , is sent to punxsutawney , p . a . to report on groundhog day . the groundhog day ceremony involves a groundhog being lifted out of a box , and if he dosen ' t see a shadow , it will be an early spring . phil really hates the ceremony , and not even his producer rita ( macdowell ) can change his mind . however , fate has a cruel trick for phil , and he starts re - living groundhog day over , and over , and over , until he gradually likes it , and rita falls in love with him . groundhog day is a well written , totally unoffensive and funny comedy . the screenwriters , director ramis and danny rubin , have written a funny , warm , but never overly senitmental comedy . although the idea of a day repeating over and over may sound tedious , there are enough good jokes to hold the audiences attention throughout the whole film . ramis ' s direction also helps , and although he dosen ' t try any flash director tricks , the film is directed well enough , and the jokes are set up well . and the editing is also good , especially when it shows one part of the day over , and over again , such as when phil tries to have the perfect night with rita . the performances are also excellent . bill murray is great fun , and his transistion from cynical to happy is smooth , and delivers his lines in his usual smary style . andie macdowell is good as rita , although sometimes she is just a little bit too sweet in some parts of the film . there is chemistry between the two leads , thankfully , otherwise the whole film would probably fall part . chris elliot , as the cameraman larry , is also funny , although you have to like his goofy style , otherwise you are really going to hate him throughout this film , and it will lower your enjoyment of the film overall . the supporting cast are n ' t bad either , with stephen tobolowsky hillarous as phils old school mate ned ryanson , and even director ramis popping up as a neurologist . in fact , there is not one dud performance in this film , and even the groundhog gets a funny scene involving a car chase . overall , there is really nothing wrong with groundhog day at all . [SEP]

### Ours-H

[CLS] in this good natured , pleasent and easy going comedy , bill murray ( ghostbusters , 1984 ) plays grumpy weatherman phil conners , who , every year , is sent to punxsutawney , p . a . to report on groundhog day . the groundhog day ceremony involves a groundhog being lifted out of a box , and if he dosen ' t see a shadow , it will be an early spring . phil really hates the ceremony , and not even his producer rita ( macdowell ) can change his mind . however , fate has a cruel trick for phil , and he starts re - living groundhog day over , and over , and over , until he gradually likes it , and rita falls in love with him . groundhog day is a well written , totally unoffensive and funny comedy . the screenwriters , director ramis and danny rubin , have written a funny , warm , but never overly senitmental comedy . although the idea of a day repeating over and over may sound tedious , there are enough good jokes to hold the audiences attention throughout the whole film . ramis ' s direction also helps , and although he dosen ' t try any flash director tricks , the film is directed well enough , and the jokes are set up well . and the editing is also good , especially when it shows one part of the day over , and over again , such as when phil tries to have the perfect night with rita . the performances are also excellent . bill murray is great fun , and his transistion from cynical to happy is smooth , and delivers his lines in his usual smary style . andie macdowell is good as rita , although sometimes she is just a little bit too sweet in some parts of the film . there is chemistry between the two leads , thankfully , otherwise the whole film would probably fall part . chris elliot , as the cameraman larry , is also funny , although you have to like his goofy style , otherwise you are really going to hate him throughout this film , and it will lower your enjoyment of the film overall . the supporting cast are n ' t bad either , with stephen tobolowsky hillarous as phils old school mate ned ryanson , and even director ramis popping up as a neurologist . in fact , there is not one dud performance in this film , and even the groundhog gets a funny scene involving a car chase . overall , there is really nothing wrong with groundhog day at all . [SEP]

### Ours-T

[CLS] in this good natured , pleasent and easy going comedy , bill murray ( ghostbusters , 1984 ) plays grumpy weatherman phil conners , who , every year , is sent to punxsutawney , p . a . to report on groundhog day . the groundhog day ceremony involves a groundhog being lifted out of a box , and if he dosen ' t see a shadow , it will be an early spring . phil really hates the ceremony , and not even his producer rita ( macdowell ) can change his mind . however , fate has a cruel trick for phil , and he starts re - living groundhog day over , and over , and over , until he gradually likes it , and rita falls in love with him . groundhog day is a well written , totally unoffensive and funny comedy . the screenwriters , director ramis and danny rubin , have written a funny , warm , but never overly senitmental comedy . although the idea of a day repeating over and over may sound tedious , there are enough good jokes to hold the audiences attention throughout the whole film . ramis ' s direction also helps , and although he dosen ' t try any flash director tricks , the film is directed well enough , and the jokes are set up well . and the editing is also good , especially when it shows one part of the day over , and over again , such as when phil tries to have the perfect night with rita . the performances are also excellent . bill murray is great fun , and his transistion from cynical to happy is smooth , and delivers his lines in his usual smary style . andie macdowell is good as rita , although sometimes she is just a little bit too sweet in some parts of the film . there is chemistry between the two leads , thankfully , otherwise the whole film would probably fall part . chris elliot , as the cameraman larry , is also funny , although you have to like his goofy style , otherwise you are really going to hate him throughout this film , and it will lower your enjoyment of the film overall . the supporting cast are n ' t bad either , with stephen tobolowsky hillarous as phils old school mate ned ryanson , and even director ramis popping up as a neurologist . in fact , there is not one dud performance in this film , and even the groundhog gets a funny scene involving a car chase . overall , there is really nothing wrong with groundhog day at all . [SEP]

Figure 14: BERT visualizations for a sample with positive emotion in Movie Reviews Dataset. A darker color represents a greater token attribution during the prediction.

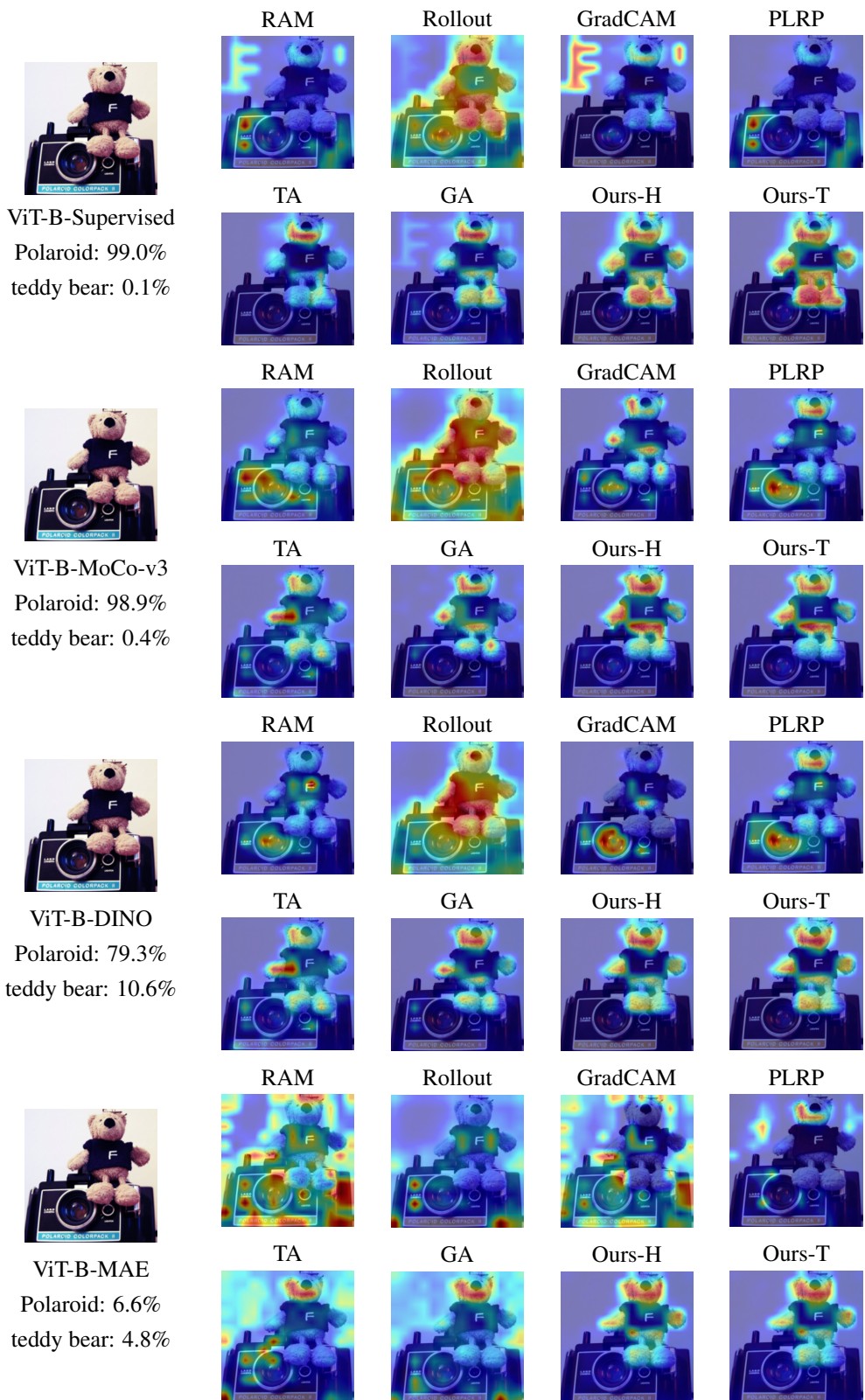

Figure 15: Visualizations *class: teddy, teddy bear* for ViT-Base with different self-supervised learning (SSL) strategies: MoCo-v3 (Chen et al., 2021), DINO (Caron et al., 2021) and MAE (He et al., 2022). Although SSL can influence explanations, our methods show stable performances and good consistency with models' properties.

