# OpenReview forum: "Beyond Intuition: Rethinking Token Attributions inside Transformers"
_TMLR — Accepted by TMLR_

### Review · Reviewer_Ut92 · 2022-11-04

**Summary Of Contributions:**

Explanation methods for transformers have been developed rapidly. Directly applying the methods used for convolutional neural networks (i.e., gradient-based methods) to transformers would yield noisier interpretations due to its complex global receptive field (multi-head self-attention) and vanishing/saturating gradients. Attention-based methods attract the community as it is intuitively linked with the attention blocks in transformers. This paper provides a more grounded exploitation of attention blocks than previous methods [1,2,3] do, each of which relies on some intuitions for summarizing the explanation. The proposed method factors the explanation into attention perception ($P$) and reasoning feedback ($F$). $P$ resembles [1]’s attention rollout but with a weaker assumption on the attention block’s output projection $W$. Index gymnastics is used to weaken the assumption of $W = I$, resulting in token-wise and head-wise attention maps. $F$ uses the integrated gradient method to focus on relevant positions in attention maps, making the method class-specific. Trustworthiness evaluations, including perturbation, segmentation, and language reasoning tests, are done and show the proposed method’s efficacy to its baselines.

[1] Abnar, Samira, and Willem Zuidema. "Quantifying Attention Flow in Transformers." Proceedings of the 58th Annual Meeting of the Association for Computational Linguistics. 2020.
[2] Chefer, Hila, Shir Gur, and Lior Wolf. "Transformer interpretability beyond attention visualization." Proceedings of the IEEE/CVF Conference on Computer Vision and Pattern Recognition. 2021.
[3] Chefer, Hila, Shir Gur, and Lior Wolf. "Generic attention-model explainability for interpreting bi-modal and encoder-decoder transformers." Proceedings of the IEEE/CVF International Conference on Computer Vision. 2021.

**Audience:**

Yes

**Broader Impact Concerns:**

I don’t have any concerns about the ethical implications of the work.

**Claims And Evidence:**

Yes

**Requested Changes:**

Please address questions in the Strengths And Weaknesses.

**Strengths And Weaknesses:**

- (S1) The authors proposed attention perception $P$ by directly expanding input-output relations of transformers using the change of basis, which is a novel way of getting token attributions not being investigated in previous studies.
- (Q1) In equation 10, the authors used $(\cdot)^{+}$ to reflect the ReLU/GeLU’s function of eliminating negative feature maps. But the implementation of the transformer’s MLP block inserts the activation between $W_{MLP, 1}$ and $W_{MLP, 2}$. Is it OK to apply a positive clamp only at the end?
- (S2) The method is evaluated not only on fully-supervised ViT/BERT but also on contrastive language-image pre-trained (CLIP) ViT and self-supervised ViT (MAE).
  - (Q2) I understand the authors focused more on the pooling (GAP) of MAE. Though I think self-supervised learning itself brings some differences in the explanation result. Examining other SSL training methods for ViT, including MoCo-v3 (contrastive) [4] and DINO (EMA distillation) [5], would clear this question. Especially, I am curious about DINO’s result as it already showed promising localization properties.
- (Q3) Which $\mathcal{L}$ is used for the visualization of CLIP? I failed to find it in the manuscript.
- (Q4) Why is CIFAR-100 used for the perturbation test of CLIP? As other image perturbation tests are done on ImageNet, it feels natural to do so for CLIP as well.
- (Typo, on page 12) Table 8 → Figure 8

[4] Chen, Xinlei, Saining Xie, and Kaiming He. "An empirical study of training self-supervised vision transformers." Proceedings of the IEEE/CVF International Conference on Computer Vision. 2021.
[5] Caron, Mathilde, et al. "Emerging properties in self-supervised vision transformers." Proceedings of the IEEE/CVF International Conference on Computer Vision. 2021.

---

> ### Author Response · Authors · 2022-11-08
> **Response (2/2) (Updated: add CLIP results on ImageNet)**
>
> **Q3&Q4 Implementation details for CLIP**
>
> For the visualization of CLIP, the $\mathcal{L}$ used is the cosine similarities between text and image. We calculate the attention and attention gradients in the image encoder to obtain the visualisations.
>
> For the perturbation test, the CLIP is used for the zero-shot classification with a designed prompt "a photo of (class name)". The current implementation of CLIP is not GPU-memory friendly. The memory demand increases almost linearly with the number of classes (or texts). We conduct the tests with CIFAR-100 which has 100 classes and yet demands 25.6G of GPU memory for calculating. Our methods are also validated on CIFAR-10 with 10 classes, whose memory usage is 3.2GB. We are working on improving the CLIP implementation to handle the ImageNet with 1000 classes. Results will be updated once the experiments are finished.
>
> *We will fix the typo and add the above discussions during revision. Thanks for the in-depth comments, and we are also very excited for further discussion if there are more questions.*
>
>
> -------
> **Updated on 24th Nov. 2022:**
> Please find below the results of perturbation tests for CLIP on ImageNet, where the results are consistent with the CIFAR-100 results and our methods still outperform the other baselines. We will update that results in the revised version.
>
> |          |           |  RAM  | Rollout | GradCAM |   GA  |   Ours-H  |   Ours-T  |
> |:--------:|:---------:|:-----:|:-------:|:-------:|:-----:|:---------:|:---------:|
> | Positive | Predicted | 14.11 |  19.32  |  21.49  | 12.78 | **12.29** |   12.32   |
> |          |   Target  |   -   |    -    |  21.34  | 11.91 |   11.55   | **11.48** |
> | Negative | Predicted | 34.02 |  28.83  |  28.78  | 35.38 | **35.74** |   35.69   |
> |          |   Target  |   -   |    -    |  28.88  | 37.09 | **37.21** |   37.10   |

---

> > ### Comment · Reviewer_Ut92 · 2022-11-19
> > **I'm not able to see response (1/2)**
> >
> > I believe the authors had responded to my Q1 and Q2, though I can only see the second response (2/2).
> > Is OpenReview malfunctioning? or Am I doing something wrong?
> >
> > Sorry for the inconvenience, but could you please re-post the first part of the response again?

---

> > > ### Author Response · Authors · 2022-11-20
> > > **Sorry for the inconvenience.**
> > >
> > > We have modified the response and we hope that you can read it this time.
> > >
> > > Really sorry for the inconvenience and thanks a lot for letting us know. We appreciate it !

---

> > ### Author Response · Authors · 2022-11-24
> > **Just a Kind Reminder**
> >
> > We find that there's no email notification when editing the previous response, but we have added the CLIP results on the previous response, with the title **Response (2/2) (Updated: add CLIP results on ImageNet)**. Please check it out. Thanks!

---

> ### Author Response · Authors · 2022-11-08
> **Response (1/2)**
>
> *Thanks for the positive comments about the novelty and contributions of this work. Following are the answers to the questions in Strength and Weaknesses.*
>
> **Q1 Is it OK to apply a positive clamp only at the end ?**
>
> In this work, in order to simplify the derivations, we made an assumption that ReLU/GeLU results in an elimination of the negative feature maps, which is also a conclusion adopted in the previous works[1,2]. Since all the terms in our attention perception, including the attentions, head-wise coefficients, and token-wise coefficients, are positive, we only take into account the negative components in Reasoning Feedback. We conduct an approximation test for the attention perception (Figure 3), where the results show the derivations under these assumptions do bring benefits to approximate the token interactions. But this is a very interesting point and we will tackle that in our future work.
>
> [1] Chefer, Hila, Shir Gur, and Lior Wolf. "Transformer interpretability beyond attention visualization." Proceedings of the IEEE/CVF Conference on Computer Vision and Pattern Recognition. 2021.
>
> [2] Chefer, Hila, Shir Gur, and Lior Wolf. "Generic attention-model explainability for interpreting bi-modal and encoder-decoder transformers." Proceedings of the IEEE/CVF International Conference on Computer Vision. 2021.
>
>
> **Q2 Other SSL training method for ViT ?**
>
> In this work, we adopt the MAE setting not only for testing the applicability of our methods to the GAP model but also for coping with the OOD problem in the perturbation test. It is questioned, e.g. by [3,4], in the perturbation test if the pixel-removed input can be out-of-distribution to models, causing the misalignment between the trustworthiness of explanations and the quantitative results. The strategy of masking and reconstructing in MAE can alleviate this issue and make the quantitative results in the perturbation test more convincing.
>
> [3]	Peter Hase, Harry Xie, Mohit Bansal. "The Out-of-Distribution Problem in Explainability and Search Methods for Feature Importance Explanations." NeurIPS 2021: 3650-3666.
>
> [4] Sara Hooker, Dumitru Erhan, Pieter-Jan Kindermans, Been Kim. "A Benchmark for Interpretability Methods in Deep Neural Networks." NeurIPS 2019: 9734-9745.
>
> For DINO, resulting from the special designed SSL, the raw attention map (RAM) method is already promising in capturing semantics. But we showed in our experiments that RAM can not totally reflect the token attributions for Supervised and MAE ViTs. And it is not class-specific either, that's to say, when it appears multiple objects on the image, the explanation can be ambiguous. We have uploaded the visualizations of ViT-DINO to the website of [zenodo](https://zenodo.org/record/7302835).
>
> We agree that the SSL can bring differences in models' explainability, however, our work focuses on providing a trustworthy explanation method to all multi-head self-attention models. We conduct additional segmentation tests with RAM and our methods for ViT-DINO. The below results demonstrate that our methods are still better in localizing the semantics. It is also surprising that the results of our methods for DINO is almost the same as Supervised ViT-Base (Table 2). This may indicate that there are some hidden and hard-to-get semantics in the supervised ViT-Base that are equal to ViT-DINO. And our methods succeed in revealing the hidden information and providing trustworthy explanations to all Transformer-based models.
>
>
> |          |     | Raw Attention Map | Ours-H    | Ours-T |
> |----------|-----|---------------|-----------|--------|
> | ViT-DINO | IoU |         65.79 | **67.42** |  66.42 |
> |          | Acc |         82.08 | **83.71** |  83.09 |
> |          | mAP |         86.79 | **87.60** |  87.26 |
>
>
> Besides, we will complete Section C in the appendix providing more visualizations with different SSL training methods for ViT, including MoCo-v3, MAE and DINO, to show that our methods can be a powerful tool in understanding different models.

---

### Review · Reviewer_PzQa · 2022-11-08

**Summary Of Contributions:**

This paper proposes a novel method to analyze the transformers by inspecting the token contributions inside transformers. It contains a head-wise and token-wise process to inspect the transformers. Extensive experiments on vision transformers, language transformers, and multi-modality transformers.

**Audience:**

Yes

**Broader Impact Concerns:**

A broader impact statement should be presented.

**Claims And Evidence:**

Yes

**Requested Changes:**

See the weakness

**Strengths And Weaknesses:**

Strength:

1. This paper proposes a novel method for inspecting transformers. Extensive experiments are conducted to demonstrate its efficiency.

2. It provides a detailed derivation process to show how to calculate the inspection and provides the code. I appreciate the authors’ efforts.

3. The visualization is fantastic for visually demonstrating the benefits of the proposed method compared with the previous inspection methods.

Weakness:

1. To evaluate the proposed methods’ performance, 5,000 images from the ImageNet val set are selected. Is there any principle behind the selection? For example, are there any special designs to make the selection balance regards the species?

2. It will be better to provide the inspection results on BERT, regarding which word is emphasized. Current visualization only focuses on the visual domain.

3. As shown in the results, the token-wise and head-wise inspection methods have a different emphases. Can the authors comment on the reasons for the difference between these two manners? It will be better to provide guidance on the selection of the two manners.

---

> ### Author Response · Authors · 2022-11-12
> **Response**
>
> *Thanks for the positive comments about the novelty and efficiency of our proposed methods. As for the questions in Weakness Section, the answers are the following.*
>
> **W1 The selection of 5,000 images**
>
> In our paper, the 5000 images are randomly selected from the validation set of ImageNet with a fixed random seed. After looking into the number of images across the total 1000 classes, we found that there do exist a slight imbalance between species. Therefore, we modify the selection by randomly choosing 5 images in each class, record the corresponding indices and conduct the additional perturbation tests with these selected images. Following are the results with class-balanced samples:
>
> |           |          |           | RAM   | Rollout | CAM   | PLRP  | GA    | TA    | Ours-H    | Ours-T    |
> |-----------|----------|-----------|-------|---------|-------|-------|-------|-------|-----------|-----------|
> | ViT-Base  | Positive | Predicted | 24.51 |   20.45 | 34.61 | 20.18 | 20.95 | 17.37 |     15.97 | **15.89** |
> |           |          | Target    |       |         | 34.07 | 20.20 | 20.01 | 16.65 |     15.14 | **15.13** |
> |           | Negative | Predicted | 45.65 |   53.86 | 41.95 | 50.62 | 48.95 | 54.53 |     57.13 | **57.97** |
> |           |          | Target    |       |         | 42.46 | 50.64 | 51.23 | 55.70 |     58.54 | **59.24** |
> | ViT-Large | Positive | Predicted | 27.71 |   21.91 | 45.66 | 30.19 | 19.21 | 20.41 |     18.01 | **17.99** |
> |           |          | Target    |       |         | 45.49 | 30.19 | 18.53 | 19.93 |     17.80 | **17.57** |
> |           | Negative | Predicted | 40.99 |   53.44 | 47.58 | 37.14 | 54.72 | 52.67 |     55.86 | **56.44** |
> |           |          | Target    |       |         | 47.69 | 37.16 | 56.13 | 53.25 |     56.33 | **56.88** |
> | ViT-MAE   | Positive | Predicted | 38.56 |   38.20 | 56.79 | 26.59 | 33.11 | 34.00 | **20.57** |     20.72 |
> |           |          | Target    |       |         | 57.24 | 26.57 | 32.63 | 33.74 | **20.22** |     20.30 |
> |           | Negative | Predicted | 40.79 |   52.03 | 24.80 | 55.34 | 57.67 | 56.92 | **65.33** |     64.48 |
> |           |          | Target    |       |         | 24.41 | 55.37 | 58.80 | 57.31 | **66.06** |     65.18 |
>
> We can see that the results of all methods on class-balanced samples are basically consistent to those on randomly selected samples, except GA on ViT-Base and TA on ViT-Large，which may result from their shifting performances across classes. Nevertheless, the results of our methods are quite stable and do not vary a lot. They still outperform the other baseline methods for all Transformer-based models.
>
> **W2 BERT visualizations**
>
> We will further add the Bert Visualization (already uploaded anonymously on website [zenodo](https://zenodo.org/record/7312901)) to the appendix during the revision. Thanks for pointing this out.
>
> **W3 Between token-wise and head-wise attention perceptions**
>
> We first describe the differences between these two manners. Compared to the Attention Rollout with the assumption $W^{(l)}=I$, token-wise and head-wise attention perceptions both consider the effects of $W^{(l)}$ but with different strategies. The token-wise manner credits its effects to the differences in the norms of token vectors while the head-wise one considers the head differences, as described in Page 6. Besides, they also differ in the implementation difficulties. For token-wise, $ZW$ is not a direct output of any component in the neural network. It may involve extra modifications to compute $ZW$ during the implementation. As for the head-wise one, it has no such issue because it only demands the attention and attention gradients for calculation.
>
> There is no strong preference between these two methods because their results are pretty similar. Their visualizations are almost consistent despite a little difference in emphasis, which is likely due to the different exactness of these two approximations. The quantitative results also demonstrate that the difference between these two manners is very limited and that they both outperform all the baseline methods, which contributes to the methodologies behind these two methods.
>
> In summary, the token-wise and head-wise expressions are both derived within Proposition 1&2, but with different approximation strategies to the same term  $W^{(l)}$. We believe that there also exist other approximation strategies besides token-wise and head-wise manners. And that's also what we will tackle in our future work.
>
> *Thanks for the very constructive comments to our work. We will add the above discussions during revision. We are also excited for further discussions if there are more questions.*

---

### Review · Reviewer_yn1D · 2022-11-20

**Summary Of Contributions:**

This paper proposes an approach to compute the amount of token attributions to the final output in Transformer models. Concretely, the amount of attribution is expressed as a Hadamard product between *attention perception* and *reasoning feedback*. *Attention perception* indicates how a token contributes to the last [CLS] representation, via unfolding each attention block with the chain rule. *Reasoning feedback* indicates how a token contribute to the final prediction based on [CLS], via back-propagating the final prediction w.r.t the last layer's attention map. The paper leverages Integrated Gradient to reduce the noise from irrelevant tokens in the *reasoning feedback* process.

The effectiveness of the resulting token attributions are validated through a series of tests:
- In the **Perturbation Test**, tokens are gradually masked based on the predicted attribution score. A sharp performance drop after masking tokens would suggest that the predicted attributions have successfully recognized important tokens.
- In the **Segmentation Test**, a semantic segmentation map is obtained from (binarized) token attribution maps. A high segmentation metric evaluated from attribution maps would suggest that the predicted attributions have successfully recognized semantically significant tokens.
- In the **Language Reasoning Test**, tokens with high attribution scores are extracted as a "rationale" supporting a classification decidion. A high F1 score between the ground-truth rationale and the rationale predicted from attribution scores would suggest the effectiveness of predicted attributions.

The paper shows that their token attributions consistently outperforms previous explanation methods on all three tests. Also, the proposed approach remains to be effective when different model sizes, modalities and [CLS]-pooling methods are used. The results hold promising for further downstream usage of token attributions for debugging, evaluating and designing Transformer models.

**Audience:**

Yes

**Broader Impact Concerns:**

There might be a dual use issue when the proposed explanation approach is applied to tasks that involve predicting human attributes. On the positive side, the explanation approach can help figure out tokens having large impacts on the model prediction, thereby increasing transparency. On the flip side, this might provide a malicious user with access to how to trick a model into making certain predictions via altering the input tokens.



**Claims And Evidence:**

Yes

**Requested Changes:**

I'd like to propose a few adjustments regarding to formatting. They are not critical to securing my recommendation for acceptance because I already believe this is a strong submission.

Section 4.1: Currently the format would mislead a reader into believing that "Faithfulness Evaluation" and "Perturbation Tests" are parallel to each other. I'd suggest put the three types of tests in a list, aligning the test names.

Section 4.2: I might be missing something, but the caption of Table1&3 did not explain what "Predicted" and "Target" correspond to.

Section 4.4: In Table4, I suppose "&check;" in the feedback column represents the proposed approach while "$G^{(L)}$" represents the old way of doing attention gradients without integration. To make the table more readable, it would be clearer to add a sentence clarifying this difference.



**Strengths And Weaknesses:**

Strengths
- It seems that the two parts in the proposed token attribution expression, *attention perception* and *reasoning feedback*, are analogous to the attributions indicated by bottom-up and top-down signals, respectively. Therefore, it seems that the proposed method well characterizes both how a token contributes to the construction of the [CLS] representation and how a token influences the final prediction.
- Since not all information stored in the final [CLS] representation will be used for a task-specific prediction, it makes sense that *reasoning feedback* helps reduce noises that could exist when only bottom-up attention attributions were considered.
- The breadth of both the test types and the model types studied in this paper assures a strong faithfulness evaluation.

Weaknesses
- The mathematical notations in Section3 are a bit hard to read. Proposition3 is referenced by Section3.1 before it is introduced at a later page.
- The effectiveness of the proposed approach is only demonstrated through explaining the [CLS] token and how [CLS] is converted to a final prediction. How the proposed approach would perform on explaining normal text tokens remains to be investigated.

---

> ### Author Response · Authors · 2022-11-24
> **Response**
>
>
> *Thanks a lot for the very positive comments about our work. We are encouraged to hear that. Concerning the weaknesses and requested changes mentioned in the review, we have the following responses:*
>
> **W1 Revision to Section 3**
>
> To cope with the numerous mathematical notations and equations, we plan to make some changes to Section 3. First, we will add a notation table after the preliminaries at the beginning of this section. Then it will be followed by an explanation of our flow of idea to make the section3 more readable. The main description about the flow of idea is as follows:
>
> Actually, we start with *Token Attribution inside Transformer* (i.e., subsection 3.1), where we introduce the main formula of our approach $\mathcal{T} = \left(\mathbf{P}^{(L)} \odot \mathbf{F}^c \right)_\mathcal{P}$. Then in subection 3.2 and 3.3, we introduce the *Attention Perception* $\mathbf{P}^{(L)}$ and the *Reasoning Feedback* $\mathbf{F}^c$ respectively. Note that, to derive the main formula in subsection 3.1, it requires the change of basis vectors, which are derived and identified in subsection 3.2. For a better presentation, we first introduce the overall framework and then the components.
>
> We hope that with a brief summary of our whole idea at the beginning, Section 3 can be easier to follow.
>
> **W2 Explanation to text tokens**
>
> In our work, we demonstrate our approach within the self-attention scope, which is applicable to both text tokens and image patches. For other NLP tasks such as text generation or machine translation tasks with a decoder architecture, our approach is not directly applicable. As addressed in Conclusion, we will tackle that in future work.
>
> Besides [CLS] pooling, we mentioned in the text (paragraph below Proposition 1) that other pooling strategies can be easily reached with same methodology. In the experiments, we also test our methods with global pooling type ViT-MAE, where our methods outperform the other baselines with a great margin.
>
> **Formatting changes**
>
> The requested changes are definitely helpful to our work. We will take them into account during the revision. Thanks a lot for pointing them out.
>
> *We will make the above changes to our work in the revised version. Thanks for the very helpful comments, and we are also excited for further discussion if there are more questions.*

---

### Review · Reviewer_Sq8c · 2022-11-22

**Summary Of Contributions:**

This paper proposes a novel method to attribute tokens in transformers. The method generates what is called attention perception and reasoning feedback combines them to get a score to interpret a transformer's decision. The design is mathematically motivated and the intuition is presented in Section 3 (which can still be improved as described later). The experimental results are impressive showing big improvements in NLP and CV domains with various tasks. Overall, I think the paper is definitely interesting to the TMLR audience and there is substantial evidence for its performance that make the paper publishable. In what follows, I will provide major points that require a revision, especially in Section 3.

**Audience:**

Yes

**Claims And Evidence:**

Yes

**Requested Changes:**

Given the effectiveness of the method and the great results, I think there is merit in this work for publication. However, major revisions are needed IMO as detailed below:

* Please rewrite Section 3; simplify notations, perhaps introduce a notations table; and make clear the flow/goal of each subsection so that it would be more understandable. Apart from this major requests, Section 3 suffers from minor issues, such as Proposition 3 is referenced before being presented, etc, which should be addressed in a revision.

* Please either remove the use of "proposition" in this section (as none of such results is rigorously provable) or add rigor to this section.

* Please add more discussion on the head-wise and token-wise variants and provide more guidance in what situations either would be preferable.

**Strengths And Weaknesses:**

Strengths:
* The design of the method is methodical and mathematically motivated.
* The experiments cover a broad range of problems in vision and natural language processing domains, with impressive performance.

Weaknesses:
* Section 3 is almost impossible to read. The introduced notation is dense and confusing; the flow of the ideas is not clear; and I was not able to understand this section despite trying to read it a few times.
* Section 3 lacks rigor. For example, a mathematical "proposition" should be provable in a rigorous manner. A proposition that says $x\approx y$ needs to at least define the sense at which $\approx$ is defined; and prove the statement.
* I was not able to understand where the differences between the head-wise and token-wise variants of the method come from. Understanding these two designs a bit more would have been preferred.

---

> ### Author Response · Authors · 2022-11-24
> **Response**
>
> *Thanks a lot for the positive comments about the contributions of our work. As for the Weaknesses mentioned, we are very thankful for the constructive comments and correspondingly we will make the following changes in our revised version.*
>
> **Readability Issue of Section 3**
>
> Thanks for the suggestions concerning the changes to Section 3. As suggested, we will revise our work correspondingly. First, we will add a notation table after the preliminaries at the beginning of this section. Then it will be followed by an explanation of the flow of our idea to make the section more understandable. The main description about the flow of idea is as follows:
>
> Actually, we start with *Token Attribution inside Transformer* (i.e., subsection 3.1), where we introduce the main formula of our approach $\mathcal{T} = \left(\mathbf{P}^{(L)} \odot \mathbf{F}^c \right)_\mathcal{P}$. Then in subection 3.2 and 3.3, we introduce the *Attention Perception* $\mathbf{P}^{(L)}$ and the *Reasoning Feedback* $\mathbf{F}^c$ respectively. Note that, to derive the main formula in subsection 3.1, it requires the change of basis vectors, which are derived and identified in subsection 3.2. For a better presentation, we first introduce the overall framework and then the components.
>
> We hope that with a brief summary of our whole idea at the beginning, Section 3 can be easier to follow.
>
> **Rigorousness**
>
> As for the rigorousness, after re-reading Section 3, we think that each approximation $\approx$ we make is stated and explained in the context. The writing might be unclear for some places. Therefore, we will rephrase the related sentences during revision. Besides, in order to know if our approximations are within the scope, we conducted an approximation test (Figure 3), where the results show that compared other baseline, the derivations under these assumptions are acceptable for approximating the token interactions.
>
> **Token-wise and head-wise attention perceptions**
>
> Concerning the differences between these two manners, unlike the Attention Rollout with the assumption $W^{(l)}=I$, token-wise and head-wise attention perceptions both consider the effects of $W^{(l)}$ but with different strategies. The token-wise manner credits its effects to the differences in the norms of token vectors while the head-wise one considers the head differences, as described in Page 6. Besides, they also differ in the implementation difficulties. For token-wise, $ZW$ is not a direct output of any component in the neural network. It may involve extra modifications to compute $ZW$ during the implementation. As for the head-wise one, it has no such issue because it only demands the attention and attention gradients for calculation.
>
> The differences between these two methods are limited , since they are both derived within Proposition 1&2. The quantitative results and visualizations also demonstrate that their explanation results are very similar and that they both outperform all the baseline methods, which contributes to the methodologies behind these two methods.
>
> In summary, the token-wise and head-wise expressions differ in the approximation strategies to the same term  $W^{(l)}$. We believe that there also exist other approximation strategies besides token-wise and head-wise manners. And that's also what we will tackle in our future work.
>
> *We will make the above changes to our work and add the discussions in the revised version. We are also excited for further discussion if there are more questions.*

---

> > ### Comment · Reviewer_Sq8c · 2022-11-24
> > **A quick note on rigor**
> >
> > Thanks for the quick response. I look forward to seeing your revisions.
> >
> > A quick note on mathematical rigor: a mathematical proposition should be provable by anyone as stated. It’s not acceptable to explain the sense at which approximation is meant in the proof or subsequent derivations. If that’s not the intended use of a preposition in this paper, then please drop the title proposition and make these statements “approximations” with proof replaced with “justification”.

---

### Author Response · Authors · 2022-11-28
**Revision**

We thank you for all the reviews concerning our work. Correspondingly we made the following changes in our revised version:

- (Page 4, Section 3) We added a brief introduction at the beginning of Section 3 to present the organization of our following paragraphs.
- (Page 4, Section 3) We added a table of notations to facilitate the reading.
- (Page 6, Sections 3.1 and 3.2) We changed the "Proposition" and "Proof" with "Approximation" and "Justification".
- (Page 8, Section 4.1) We aligned and itemized the three tests in Faithfulness Evaluation.
- (Page 9, Section 4.1) We added the explanations to "Predicted" and "Target".
- (Page 10, Table 2, Section 4.2) We changed the results of perturbation test with the balanced samples and modified the corresponding writing in ViT section.
- (Page 12, Table 4, Section 4.2) We changed the dataset in CLIP perturbation test from CIFAR100 to ImageNet and reported the obtained results.
- (Page 13, Section 4.3) We mentioned the OOD issue in perturbation test and the advantage of using MAE in ViT-MAE section.
- (Page 13, Table 5, Section 4.4) We changed the notation from $\checkmark$ to $\int \mathbf{G^{(L)}}$ to highlight the integration and changed the results of perturbation test with balanced samples.
- (Page 13) We added Section 4.5 to discuss the differences between our token-wise and head-wise manners.
- (Page 14) We added the Boarder Impact Statement.
- (Page 20, Appendix C) We added a discussion section about the influences of SSL in explanation.
- (Pages 24&25, Appendix D) We added the BERT visualizations in Appendix D.
- (Page 26, Appendix D) We added visualizations of ViTs with different SSL methods in Appendix D.

Please check our revised version and we are also excited for further discussions if there are any questions.

---

### Author Response · Authors · 2023-01-09
**Updated Version**

*We updated a new revised version of our work with the following changes:*

- (Page 3, Figure 2) As suggested, we modified Figure 2 with a caricature of Transformers identifying the notations we use in the following derivations. Besides, we added in the left a roadmap of our following paragraphs, which illustrates our method briefly.
- (Page 18, Appendix A) We moved the table of notations to the Section A of Appendix.
- (Page 4, Section 3) With the above changes, we rephrased the sentences at the beginning of Section 3.
- (Page 7, Section 3) We added a paragraph at the end of Section 3, summarizing the contents of above sub-sections and highlighting the key points of our methods.


*Thanks a lot for all the useful suggestions to our work and we hope that the above changes can alleviate the hard-to-read problem of Section 3.*

---

### Author Response · Authors · 2023-01-31
**Camera Ready Revision**

*We thank AE and all the reviewers for their positive comments and constructive suggestions to our work. Taking into account the mentioned points about Section 3, we uploaded a camera ready revision with the following changes:*

- (Page 3, Section 3) We reorganized the paragraph at the beginning of Section 3 with the section numbers and equations for a closer connection to the roadmap in Figure 2.
- (Page 3, Section 3, Preliminaries) We added a paragraph following Transformers to recall the important mathematical concepts in our derivation such as basis, coordinates and partial derivatives, and to introduce their notations such as $B$ and $P_{0, \cdot}^{(L)}$.
- (Page 3-7, Section 3.1 & Section 3.2) We rephrased some sentences in Section 3 to avoid the confusing pointer and reference.
- (Pages 3, Section 3.1 & Appendix A) We changed the notation for coordinates from $\widetilde{x}_i$ to $t_i$ to better distinguish between the basis vector $\widetilde{X}_i$ and coordinates.
- (Page 1, footer) We provided the link to our code on Github.

---

> ### Comment · Reviewer_Sq8c · 2023-02-07
> **Thanks for the revisions!**
>
> Dear authors:
>
> Many thanks for your several rounds of revisions! I think the readability of Section 3 has significantly improved, and I hope it will help others build on your great work.
>
> Best wishes, \
> Reviewer Sq8c

---

### Decision · Action_Editors · 2022-12-18

**Recommendation:** Accept with minor revision

**Comment:**

Thanks for your time and all the revisions that you have done to this paper, the reviewers feel they have substantially strengthened the work and hopefully its final impact on the community.  Given my discussions with them, we would like to accept the paper with minor revisions.

Specifically, they have provided the following feedback:

> Section 3 would benefit from a close read by a colleague unfamiliar with the research to help with clarity.  The following are a couple of example pointers from the first paragraph of Section 3 (page 4), where statements are unclear which would likely be caught by a new reader.  There are similar small points thought-out the rest of Section 3 they will be able to help with
> - there is still a pointer to (12) that is hard to understand,
> - $\tilde{x}_i$ needs to be better defined,
> - it is not clear what $(Z_{CLS})_B$ is,
> - Approximation 2 is referenced before it is given,
> - $P_0^{(L)}$ not defined.
>
> The new roadmap picture is great! but it is not discussed in the main text. Please add a paragraph in the beginning of Section 3 that at a high level connects all of the components of the subsequent subsections that are used to obtain the final approximation, and ground that description both in Section numbers, equation numbers, and the roadmap figure for the reader to better understand where Section 3 is headed.

All the reviewers are positive on your work and results, but worry that the impact will be diminished if you lose readers during the critical methods section.


**Audience:**

The direct audiences are those interested in interpretability within NLP/CV, but given the broad adoption of transformers and the utility of well-reasoned attribution, the audience is potentially a much broader swath of the ML community.

**Claims And Evidence:**

The work focuses on a new method for token attribution within transformers with the aim to provide interpretability to a prediction. The work provides formal justification for the approach in addition to a number of experimental conditions -- results are presented in both NLP and CV.  Additionally, ViT based results are compared across initializations and sizes.  Additionally, the results are compared to appropriate canonical works.